# FlowBind: Efficient Any-to-Any Generation with Bidirectional Flows

**Yeonwoo Cha**[*]   **Semin Kim**[*]   **Jinhyeon Kwon**   **Seunghoon Hong**
KAIST
{ckdusdn03, seminkim, jh.kwon, seunghoon.hong}@kaist.ac.kr
[*] Equal contribution

## Abstract

Any-to-any generation seeks to translate between arbitrary subsets of modalities, enabling flexible cross-modal synthesis. Despite recent success, existing flow-based approaches are challenged by their inefficiency, as they require large-scale datasets often with restrictive pairing constraints, incur high computational cost from modeling joint distribution, and rely on complex multi-stage training. We propose **FlowBind**, an efficient framework for any-to-any generation. Our approach is distinguished by its simplicity: it learns a shared latent space capturing cross-modal information, with modality-specific invertible flows bridging this latent to each modality. Both components are optimized jointly under a single flow-matching objective, and at inference the invertible flows act as encoders and decoders for direct translation across modalities. By factorizing interactions through the shared latent, FlowBind naturally leverages arbitrary subsets of modalities for training, and achieves competitive generation quality while substantially reducing data requirements and computational cost. Experiments on text, image, and audio demonstrate that FlowBind attains comparable quality while requiring up to 6× fewer parameters and training 10× faster than prior methods. The project page with code is available at `https://yeonwoo378.github.io/official_flowbind`.

## 1 Introduction

Recent progress in flow-based generative models has delivered state-of-the-art performance in multi-modal generation. By conditioning on a given input modality, these models excel at specialist tasks such as text-to-image (Esser et al., 2024; Black Forest Labs, 2024) or text-to-audio synthesis (Liu et al., 2024; Huang et al., 2023), demonstrating their strength in learning continuous cross-modal transformations. However, these successes are largely confined to fixed input and output mapping, and extending flow models to support true any-to-any generation, where arbitrary subsets of modalities can be generated given any other subsets, remains an open challenge.

Bridging the gap from specialist to generalist flow models introduces fundamental hurdles, primarily due to requirements of multi-modal data and computational cost. Frameworks that rely on a central anchor modality, typically text (Tang et al., 2023), require each modality to be paired with text during training so that all modalities can be aligned through the shared text representation. This design is restrictive, as it prevents the model from learning the rich, direct correlations that exist beyond language. Conversely, methods that model the full joint conditioning of all modalities (Li et al., 2025b) can achieve expressive generation performance but at a steep cost: they require some fully-paired data for stable training, which is scarce, and their computational complexity often scales quadratically with the number of modalities. These data and compute issues render them impractical for real-world scenarios with a large and diverse set of modalities.

Beyond the computational cost, a significant hurdle for generalist models is the complexity of their training pipelines. Rather than a single, unified process, these frameworks often rely on intricate, multi-stage procedures. These stages separately optimize the encoding components for modality alignment and the decoding components responsible for the model's generative capabilities. This staged approach is evident in prominent models; for instance, CoDi (Tang et al., 2023) employs a

multi-stage process that separates modality alignment from joint generation. Similarly, OmniFlow (Li et al., 2025b) requires a distinct post-training phase after merging its core components. Such multi-stage pipelines can be brittle, difficult to optimize, and hinder the development of truly seamless, end-to-end generative models.

We introduce FlowBind, a simple flow-based model that addresses these limitations. FlowBind introduces a learnable shared latent capturing cross-modal commonality, and connects each modality to this latent through its own invertible flow. All components are trained jointly under a single flow matching objective, while the learned flows enable direct any-to-any translation at inference. Because each flow requires only its modality paired with the latent, the method naturally supports training with partially paired data while reducing computational cost. This design yields a simple, efficient, and data-flexible solution for general-purpose any-to-any generation.

In summary, our main contributions are as follows: **(1)** We introduce a flow-based framework for any-to-any generation that factorizes multi-modal interactions through a learnable shared latent, enabling training from arbitrary paired data with low computation budget. **(2)** Our method jointly optimizes both the shared latent and all modality-specific flows under a single flow-matching loss, avoiding the multi-stage pipelines. **(3)** Experiments on text, image, and audio demonstrate that FlowBind achieves competitive quality with substantially reduced data and computation compared to representative baselines, while flexibly supporting any-to-any translation.

## 2 PRELIMINARIES

**Flow Matching**  Conditional Flow Matching (Lipman et al., 2023) is a simulation-free framework for learning a continuous transformation between a source distribution $p_0$ and a target distribution $p_1$. This transformation is defined by an Ordinary Differential Equation (ODE), $\frac{dz_t}{dt} = v_\theta(z_t, t)$, where a drift network $v_\theta$ parametrizes the time-dependent vector field. With linear interpolation path $z_t = (1 - t) z_0 + t z_1$ with $(z_0, z_1) \sim (p_0, p_1)$, the target velocity is simply $z_1 - z_0$, and the objective becomes:

$$\mathcal{L}_{\text{FM}}(\theta) = \mathbb{E}\left[\|v_\theta(z_t, t) - (z_1 - z_0)\|^2\right]. \tag{1}$$

At the optimum, Eq. 1 yields the conditional expectation of the target velocity:

$$v_\theta^\star(x, t) = \mathbb{E}[z_1 - z_0 \mid z_t = x]. \tag{2}$$

Generation is then performed by integrating the learned drifts over time:

$$z_{t_1} = z_{t_0} + \int_{t_0}^{t_1} v_\theta(z_t, t)\, dt = \text{ODESolve}(z_{t_0}, v_\theta, t_0, t_1) \tag{3}$$

Note that, under standard Lipschitz conditions, the induced flow is invertible *i.e.*, the ODE can be integrated forward or backward in time to induce samples from $p_0$ or $p_1$.

**Any-to-Any Generative Flows**  The goal of any-to-any generation is to learn a unified model that can translate between arbitrary subsets of modalities. Given $N$ modalities $\mathbf{z} = (z^1, \ldots, z^N)$, this amounts to modeling their joint distribution $p(\mathbf{z})$ so that for any $S_{\text{in}}, S_{\text{out}} \subseteq \{1, \ldots, N\}$, the model can perform any-to-any generation by sampling from conditional probability $p(\mathbf{z}^{S_{\text{out}}} | \mathbf{z}^{S_{\text{in}}})$.

Existing flow-based approaches address this problem by constructing continuous trajectories that transform i.i.d. Gaussian noise $\mathbf{z}_0 \sim \pi_{\text{prior}}$ into data samples $\mathbf{z}_1 \sim \pi_{\text{data}}$. Representative examples include CoDi (Tang et al., 2023) and OmniFlow (Li et al., 2025b), which mainly differ in how they synchronize trajectories across modalities. CoDi learns modality-specific encoders that align all modalities to a shared text embedding, which then serves as the conditioning signal for per-modality denoising networks $\epsilon^i(z_t^i, t, \hat{c}^{\text{text}})$. In contrast, OmniFlow learns a time-decoupled joint velocity field $v(z_{t_1}^1, \ldots, z_{t_N}^N, t_1, \ldots, t_N)$, where the interpolation path for each modality is explicitly conditioned with the other modalities to ensure alignment.

Despite their empirical success, existing flow-based methods face several limitations. First, they cannot fully leverage arbitrary paired modalities for any-to-any generation: CoDi requires each modality to be paired with text to establish a canonical embedding, while OmniFlow relies heavily

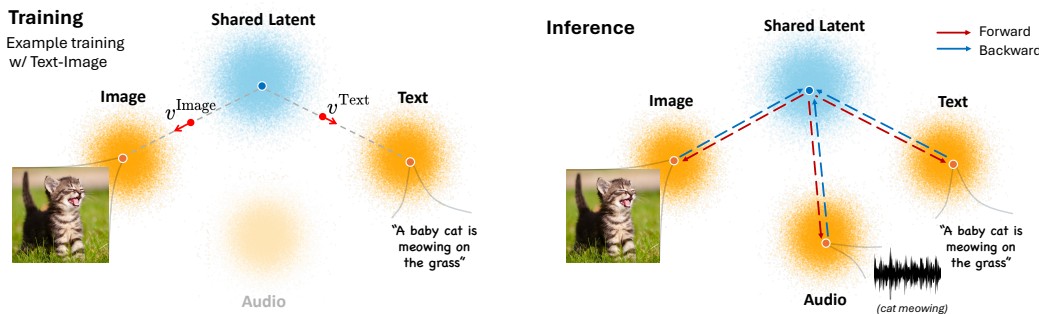

(a) **Training** both shared latent and drifts.        (b) **Inference** with per-modality drifts.

Figure 1: An overview of FlowBind. (a) During training, we jointly learn the shared latent and per-modality drift networks in a single stage. (b) At inference, the learned drift networks perform flexible any-to-any generation by solving per-modality ODEs forward and backward in time.

on fully-paired data (i.e., samples where all $N$ modalities are present) for stable training [1]. Second, both methods require multi-stage training: CoDi separately learns the shared representation and denoising networks, whereas OmniFlow pre-trains drift networks for each modality pair before joint training. Finally, they operate in high-dimensional representations, leading to substantial computational cost and slow convergence.

## 3    FLOWBIND

To address the aforementioned challenges, we propose **FlowBind**, a unified flow-based framework for any-to-any generation. FlowBind is designed to overcome key drawbacks of prior methods: it supports single unified training procedure, operates with lower computational overhead, and fully exploits partially paired data for effective learning.

The key idea of FlowBind is to replace the fixed Gaussian prior with a *learnable, shared* distribution that encapsulates common information across modalities. This acts as a latent anchor, where each modality is connected to it via their own invertible, per-modality flows (Figure 1). With this factorization, FlowBind achieves alignment across modalities naturally via the shared distribution, unlike existing approaches that anchor all modalities to text (Tang et al., 2023) or couples them through a joint drift (Li et al., 2025b). Meanwhile, both the shared distribution and the per-modality flows are learned jointly with only standard flow matching loss using partially paired data.

Formally, consider a subset of multi-modal data $\mathbf{z}^S = \{z^i | i \in S\}$ with $S \subset \{1, \ldots, N\}$, which is sampled from a joint distribution $\mathbf{z}^S \sim \pi_{\text{data}}^S$. Assume that there exists a shared latent $z^* \sim \pi_{\text{shared}}^S$ that encompasses the common information of all individual modality in $\mathbf{z}^S$. Then for each $i \in S$, FlowBind learns a straight interpolation path that bridges the data $z^i$ to the shared latent $z^*$ by:

$$z_t^i = tz^i + (1 - t)z^* \tag{4}$$

$$\frac{\partial z_t^i}{\partial t} = v^i(z_t^i, t), \tag{5}$$

where $v^i$ denotes the modality-specific velocity field. Note that multi-modal flows are factorized per modality given the shared latent (Eq. 4), and the shared latent implicitly aligns these flows across modalities (Eq. 5). During training, the shared latent is instantiated as $z^* = H_\phi(\mathbf{z}^S)$ through an auxiliary encoder $H_\phi$, whose marginal approximates $\pi_{\text{shared}}$ and is optimized jointly with the per-modality drift networks $v_{\theta^i}$ (Figure 1(a)). At inference, FlowBind relies only on the learned drift networks: owing to the invertibility of the direct flows, both inferring the shared latent from input modalities and generating outputs from the latent are achieved by a single drift network per modality (Figure 1(b)). Details of the training and inference procedures are provided in Section 3.1.

---

[1]Although partially paired data can be used for training in principle, performance and stability are reported to depend strongly on fully paired data; see Appendix B.2 of Li et al. (2025b).

Following prior works (Esser et al., 2024; Liu et al., 2024), FlowBind operates in a compressed latent space obtained by per-modality autoencoders. However, instead of low-level, high-dimensional latent, we adopt compact and semantic representations extracted by strong encoders in each modality, paired with decoders that reconstruct modality-specific details from the encoded feature. This design enables FlowBind to focus on shared structure in a low-dimensional space, making cross-modality alignment simpler and training both faster and efficient.

Taken together, FlowBind provides several advantages over existing approaches. By introducing a shared latent space, FlowBind factorizes the multi-modal flow into independent per-modality drifts, allowing them to operate in isolation with reduced computational cost. This factorization also naturally enables training with arbitrary paired modalities: since each drift network learns only to connect its modality to the shared latent, learning does not depend on specific modalities or fully-paired data. Finally, both the shared latent and modality-specific drifts are optimized jointly with a single flow matching objective, avoiding the multi-stage training pipelines of prior works and yielding a simple and efficient framework.

## 3.1 TRAINING AND INFERENCE

**Learning Objective** During training, the auxiliary encoder $H_\phi$ and the set of modality-specific drift networks $\{v_{\theta^i}\}_{i=1}^N$ are optimized jointly under the flow matching framework in Eq. 4 and 5. Given a partially paired sample $\mathbf{z}^S$, the auxiliary encoder produces a shared latent $z^* = H_\phi(\mathbf{z}^S)$, and for each modality $i \in S$, the drift network $v_{\theta^i}$ is trained to approximate the velocity field along the path between $z^i$ and $z^*$. This leads to the training objective:

$$\mathcal{L}(\theta, \phi) = \mathbb{E}_{t, \mathbf{z}^S, z^*} \left[ \sum_{i \in S} \left( \left\| v_{\theta^i}(z_t^i, t) - (z^i - z^*) \right\|^2 \right) \right], \qquad (6)$$

where $\theta = \{\theta^1, ..., \theta^N\}$. In principle, this couples the two components: drift networks learn to predict the displacement toward each modality endpoint, while the auxiliary encoder is encouraged to provide a shared latent from which every modality can be recovered to aid drift networks.

However, this formulation admits degenerate solutions. For example, if the encoder collapses to a constant output such as $z^* = 0$, the drift networks can trivially fit $v^i(z_t^i, t) = z^i$ with $z_t^i = tz^i$ and achieve the zero loss for $t \in (0, 1]$, leaving the encoder with no meaningful supervision (Kim et al., 2024). The underlying reason is that flow matching enforces transportation between two fixed endpoints but does not itself constrain the distribution of encoder outputs. Prior works on direct flow (Liu et al., 2025; He et al., 2025) address this by adding explicit regularizers, such as contrastive losses on the encoder, but these introduce additional computation, hyperparameters, and scalability bottlenecks especially with increasing number of modalities.

In contrast, we show that both stabilization and meaningful learning of the encoder can be achieved within the flow-matching objective itself. Our approach is simple: for $t \in (0, 1]$, we stop gradients through the auxiliary encoder to stably train the drift networks, while at $t=0$, the encoder is directly updated together with the drifts. Despite its simplicity, this scheme effectively prevents collapse and provides the encoder with a meaningful learning signal, as we elaborate below.

**Analysis on Encoder Objective** To understand what the auxiliary encoder learns under our training strategy, we analyze the flow matching loss at $t=0$. Substituting the Bayes-optimal drift $v^\star(z_t, t)$ (Eq. 2) into Eq. 6 at $t = 0$ gives encoder's effective objective:

$$\mathcal{L}(\phi) = \mathbb{E}\left[\| v^\star(z^*, 0) - (z^i - z^*) \|_2^2\right] = \mathbb{E}\left[\| \mathbb{E}[z^i \mid z^*] - z^i \|_2^2\right] = \mathbb{E}\left[\mathrm{Var}(z^i \mid z^*)\right]. \quad (7)$$

This shows that, at $t=0$, the encoder is explicitly optimized to minimize the conditional variance of each modality given the shared latent. The term $\mathbb{E}\left[\mathrm{Var}(z^i \mid z^*)\right]$, often referred to as the *unexplained variance*, measures how much information about modality $i$ remains outside of $z^*$. By the law of total variance (Grimmett & Stirzaker, 2001), reducing this quantity equivalently increases the explained variance of $z^i$ by $z^*$. Since the optimization is carried out jointly across all modalities, the encoder is therefore driven to shape $z^*$ so that it retains predictive information about each modality, ensuring that the shared latent becomes increasingly informative for cross-modal alignment.

More generally, when the drift networks are not optimal, Eq. 6 at $t=0$ decomposes into unexplained variance and an approximation error of the drifts:

**Proposition 1 (Equivalence at $t = 0$)** *For any parameters $(\theta, \phi)$ and modality subset $S$, the flow matching loss (Eq. 6) at $t = 0$ decomposes as follows:*

$$\mathcal{L}(\theta, \phi) = \underbrace{\sum_{i \in S} \mathbb{E}\big[\mathrm{Var}(z^i \mid z^*)\big]}_{\text{unexplained variance}} + \underbrace{\sum_{i \in S} \mathbb{E}\Big[\big\| v_{\theta^i}(z^*, 0) - \mathbb{E}[\, z^i - z^* \mid z^*] \big\|^2\Big]}_{\text{approximation error of } v_{\theta^i}}.$$

*A formal proof is provided in Appendix A.*

This decomposition reveals that even when the drift networks are imperfect, the auxiliary encoder $H_\phi$ is consistently driven to minimize unexplained conditional variance while simultaneously optimizing the shared latent to enable better drift approximation of their targets. In this way, our training strategy encourages the auxiliary encoder and the drift networks to remain tightly coupled: the drifts learn to predict each modality endpoint from the shared latent ($t \in [0, 1]$), while the encoder is driven to shape the latent into a representation from which all modalities can be reliably recovered ($t = 0$). During training, we balance the drifts and encoder training by sampling from the mixture $t \sim (1 - \alpha)\mathrm{Unif}(0, 1) + \alpha\delta(t = 0)$. The training procedure is given at Algorithm 1.

**Inference**    After training, FlowBind performs versatile any-to-any generation relying solely on the learned per-modality flows, without utilizing the auxiliary encoder. Given a source modality $i$, we first project it onto the shared latent by integrating its backward flow, and then map the shared latent to the target modality $j$ via the corresponding forward flow:

$$\hat{z}^* = \mathrm{ODESolve}(z^i, v_{\theta^i}, 1, 0), \qquad \hat{z}^j = \mathrm{ODESolve}(\hat{z}^*, v_{\theta^j}, 0, 1) \tag{8}$$

When conditioning on multiple source modalities $\mathbf{z}^S$, FlowBind obtains per-modality latent estimates $\hat{z}^{(*,i)}$ by solving the corresponding backward flows independently. These estimates are then aggregated into the shared latent $\hat{z}^*$ by simple averaging. Finally, the target modality is generated by integrating its forward flow starting from $\hat{z}^*$. The inference procedure is given at Algorithm 2.

## 4    RELATED WORK

**Any-to-Any Generation**    A prominent paradigm for any-to-any generation tokenizes all modalities into a discrete space and trains a single sequence model to predict the unified stream autoregressively. In this setup, a powerful large language model performs cross-modal sequence generation, with tokenized data of all modalities. Some works (Team, 2024; Kou et al., 2025) focus on interleaved generation solely on text-image generation, while others (Wu et al., 2024; Zhan et al., 2024) extend to broader multi-modal scenarios including speech (Wang et al., 2024b) and even robotics (Lu et al., 2024). Training such models typically involves multi-stage procedures and often relies on instruction fine-tuning, which requires datasets with detailed textual descriptions and usually depends on large language models. Consequently, text-paired data is often required for training.

Another line of work utilizes discrete diffusion models, often by adapting them to operate on discrete token spaces (Rojas et al., 2025; Shi et al., 2025). These methods, which typically focus on text-image generation tasks, leverage the high-quality synthesis capabilities of diffusion for multi-modal scenarios. For instance, UniDisc (Swerdlow et al., 2025) highlights the controllability of this approach by framing various conditional generation tasks, such as inpainting.

**Direct Flow-based Models**    Recent flow-based models have explored learning direct, data-to-data invertible mappings between two data distributions (Li et al., 2023; Wang et al., 2024a), predominantly focusing on text-image pairs (Liu et al., 2025; He et al., 2025). This approach represents a fundamental departure from traditional generative flows that typically learn bridging from fixed prior distributions (e.g., standard Gaussian) to target data distributions through conditional generation mechanisms. To facilitate these direct transformations, existing methodologies designate latent distributions of one modality (i.e., source distribution) as a learnable embedding. This is achieved by introducing an encoder for the source modality and constructing additional loss terms that align the source and target modalities, such as contrastive learning objectives.

While our approach shares foundational ideas with prior work, its emphasis and formulation differ. Existing methods typically rely on multiple loss terms to stabilize training and to optimize endpoint embeddings; in contrast, we employ a single, unified flow objective to achieve the same optimization. Moreover, we pursue direct flows for multi-modality connectivity, whereas most prior efforts have concentrated on two-modality settings, especially text–image generation.

Table 1: Comparison of computational cost. #(A-B) indicates the number of training samples for each dataset combination. Training time for CoDi is omitted due to absence of training code and details. For OmniFlow, we report the training time only for the final joint training stage.

| Model | Train Param. | GPU-hr | Number of Traning Data | | | | Joint Training |
|-------|-------------|--------|--------|--------|--------|--------|----------------|
| | | | #(T–I) | #(T–A) | #(I–A) | #(T–A–I) | |
| CoDi | 4.3B | - | 400M | 3.5M | 1.9M | - | NO |
| OmniFlow | 3.2B | 480hr* | 28M | 2.4M | - | 2.2M | NO |
| **FlowBind** | 568M | 48hr | 310K | 96K | 180K | - | YES |

## 5 EXPERIMENTS

We conduct an extensive evaluations on any-to-any generation tasks across text, image, and audio modalities, covering a wide range of input-ouput modality combinations including one-to-one, one-to-many and many-to-one generation tasks. For baselines, we mainly consider approaches on flow-based any-to-any generative modeling, namely CoDi (Tang et al., 2023) and OmniFlow (Li et al., 2025b). Qualitative results for various input-output modality combinations are available on our official project page: `https://yeonwoo378.github.io/official_flowbind`.

**Tasks and Evaluation Protocol**    For one-to-one generation, we consider all six possible tasks that consist of text, image and audio, and discuss its result in Sec. 5.2. We adopt established evaluation metrics for one-to-one generation, where standard measures of generation quality and cross-modal alignment are available. Specifically, generation quality is assessed using established modality-specific measures: FID (Heusel et al., 2017) for images, FAD (Kilgour et al., 2019) for audio, and CIDEr (Vedantam et al., 2015) for text captions. Cross-modal alignment is evaluated through pairwise similarity metrics: CLIP scores for text-image pairs (Hessel et al., 2021), CLAP scores for text-audio pairs (Elizalde et al., 2023), and Audio-Image-Similarity (AIS) (Wu et al., 2022) for image-audio pairs. Evaluations are done at held-out test set for text-audio and image-audio tasks, while we employ widely adopted zero-shot benchmark in MS-COCO for text-to-image and image-to-text tasks. Detailed evaluation protocols and descriptions of the specialist baseline models are in Appendix D.

For many-to-many generation, we conduct a qualitative analysis of more challenging settings to validate FlowBind's cross-modal capabilities. Furthermore, since no standard benchmark exists for many-to-many generation, we introduce a tailored quantitative evaluation metric and report corresponding results in Sec. 5.3, demonstrating that FlowBind consistently outperforms the baselines.

**Implementation Details**    We employ EmbeddingGemma (Team et al., 2025) for textual semantic latent, CLIP (Radford et al., 2021) for visual latent with Stable-UnCLIP (HuggingFace, 2025) as decoder, and CLAP (Elizalde et al., 2023) features for audio synthesis conditioning. Note that these modality-specific encoders and decoders are frozen during the training of FlowBind. We employ MLP-based architecture with residual connections for both auxiliary encoders and drift networks, with AdaLN-zero for time modulation (Peebles & Xie, 2023). More detailed information, including the architectural and training specifications, can be found in Appendix C.

### 5.1 INSTANTIATION OF FLOWBIND

To highlight FlowBind's training efficiency, we instantiate FlowBind as a relatively lightweight model and train it on a smaller dataset, jointly learning modality-specific drift networks and a shared latent space. We summarize the details of our instantiation of FlowBind in Table 1, making comparison to previous flow-based any-to-any generation models. Compared to baselines, FlowBind achieves any-to-any generation with considerably less computations and efforts. When comparing the computational cost, FlowBind operates on low-dimensional, compact representation space, yielding a lightweight model with less than 1B trainable parameters. This design choice makes FlowBind to be trained much faster, using about $10\times$ less compute compared to OmniFlow, in terms of GPU-hours. We also use much smaller data compared to the baselines (0.15 % of CoDi or 1.79 % of OmniFlow). Overall, these comparisons position FlowBind as a practical and scalable framework for any-to-any generation with substantially reduced compute and data budgets. In subsequent sections, we demonstrate that our efficient any-to-any generation model attains strong cross-modal generation performance.

Table 2: Fidelity assessment on one-to-one evaluation benchmarks.

| Category | Model | T → I FID ↓ | I → T CIDEr ↑ | T → A FAD ↓ | A → T CIDEr ↑ | I → A FAD ↓ | A → I FID ↓ |
|----------|-------|------|-------|------|-------|------|------|
| *Specialists* | SD3-Medium | 25.40 | – | – | – | – | – |
| | FLUX.1 | 22.06 | – | – | – | – | – |
| | LLaVA-NeXT | – | 109.3 | – | – | – | – |
| | TangoFlux | – | – | 1.41 | – | – | – |
| | AudioX | – | – | 3.09 | – | – | – |
| | Qwen2-Audio | – | – | – | 4.64 | – | – |
| | Seeing & Hearing | – | – | – | – | 5.31 | – |
| | Sound2Vision | – | – | – | – | – | 42.55 |
| *Generalists* | UnifiedIO2-L | 21.54 | 134.7* | 8.31 | 12.15 | – | – |
| | CoDi | 24.80 | 16.40 | 9.84 | 6.62 | 14.58 | 50.4 |
| | OmniFlow | 22.97 | 44.20 | 4.20 | 31.79 | 5.67 | 106.03 |
| | **FlowBind** | **17.39** | **46.26** | **4.19** | **55.11** | **2.50** | **26.60** |

Table 3: Alignment results on one-to-one evaluation benchmarks.

| Category | Model | T → I CLIP ↑ | I → T CLIP ↑ | T → A CLAP ↑ | A → T CLAP ↑ | I → A AIS ↑ | A → I AIS ↑ |
|----------|-------|------|------|------|------|------|------|
| *Specialists* | SD3-Medium | 31.60 | – | – | – | – | – |
| | FLUX.1 | 31.06 | – | – | – | – | – |
| | LLaVA-NeXT | – | 32.14 | – | – | – | – |
| | TangoFlux | – | – | 42.71 | – | – | – |
| | AudioX | – | – | 29.29 | – | – | – |
| | Qwen2-Audio | – | – | – | 17.09 | – | – |
| | Seeing & Hearing | – | – | – | – | 75.11 | – |
| | Sound2Vision | – | – | – | – | – | 62.39 |
| *Generalists* | UnifiedIO2-L | 30.71 | 30.73 | 13.48 | 18.68 | – | – |
| | CoDi | 30.26 | 26.24 | 10.79 | 17.94 | 61.55 | 74.26 |
| | OmniFlow | **31.52** | 27.71 | 24.23 | **45.08** | 71.71 | 59.22 |
| | **FlowBind** | 28.35 | **29.74** | **29.08** | 36.70 | **82.89** | **78.17** |

## 5.2 RESULTS ON ONE-TO-ONE GENERATION

**Effectiveness of FlowBind**  We demonstrate the effectiveness of FlowBind under all six pairwise one-to-one generation scenarios in Table 2 and Table 3. While the core goal of FlowBind lies on efficient modeling of any-to-any generation, we also observe the resulting model shows strong capability in cross-modal generation tasks. Compared to CoDi and OmniFlow, FlowBind achieves the best quality metrics in all six one-to-one generation tasks, while showing superior alignment score on four tasks among six. We also note that baselines such as OmniFlow are initialized from strong specialist model (*i.e.*, SD3-Medium) that excels at text-image alignment, which explains their particularly good performance on text-to-image alignment scores. We compare FlowBind with UnifiedIO2-L (Lu et al., 2024), a recent LLM-based any-to-any generation method. FlowBind is comparable on text–image tasks and stronger in other settings, suggesting that our formulation may provide benefits beyond flow-based methods. As an overall, we conclude that FlowBind shows promising performance on the evaluated one-to-one generation tasks.

An interesting observation is that FlowBind exhibits substantial gains in the image-audio generation, where it significantly outperforms among generalists and even dedicated specialist, without making modality-specific adjustments. We conjecture the impressive performance of FlowBind at audio-image correspondence stems from the introduction of learnable shared latent space, which is designed to contain meaningful information about each modality (Section 3.1) and learns directly from audio-image pair. Instead of learning a shared latent space from arbitrarily paired data, CoDi employs an text-anchored design, using only text-paired data during its multimodal alignment stage. This design choice of CoDi makes alignment between non-text modality, such as audio-image alignment, to be indirectly captured with the aid of text. We note that OmniFlow is also implicitly relying on text representation, given the fact that its weights are initialized from pretrained text-to-image and

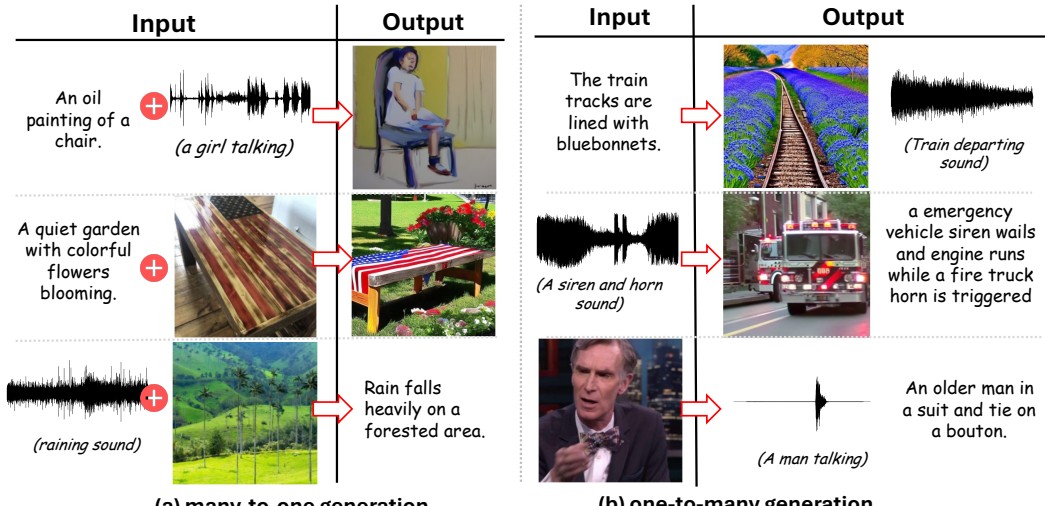

Figure 2: Qualitative results on various many-to-many generation tasks. More results and comparisons with baselines are presented in Appendix H.

text-to-audio models at the beginning of second-stage any-to-any training. In contrast, FlowBind learns a shared latent space directly from the available paired training data, treating all modalities symmetrically and thereby offering a more suitable representation for cross-modal generation.

**Train Efficiency**  While achieving promising performance relative to prior any-to-any generation models, we emphasize that FlowBind is trained with substantially less computation and effort. As previously shown in Table 1, the demonstrated strong performance of FlowBind is achieved using 6 times less training parameters and 10 times less compute compared to OmniFlow. Moreover, FlowBind employs a unified training objective, in contrast to prior works that require complex multi-stage training pipelines. Consequently, FlowBind can be trained with less effort without cumbersome hyperparameters and additional computations that emerge from those complex training procedures.

**Data Efficiency**  In terms of data efficiency, FlowBind achieves any-to-any generation with a much smaller training dataset, using 0.15 % of CoDi and 1.79 % of OmniFlow. We conjecture the training can be done with much smaller dataset because we choose to model flow between high-level representations. By doing so, the cross-modal generation capability is decomposed into inter-modal alignment and intra-modal generation in FlowBind. Our drift network is only required to capture inter-modal correspondence, as per-modality frozen encoder-decoders take charge in capturing intra-modal generative capability. This would enable FlowBind to quickly capture cross-modal alignment with smaller datasets.

## 5.3 RESULTS ON MANY-TO-MANY GENERATION

Beyond our extensive one-to-one evaluations, we further conduct qualitative analyses to assess capability of FlowBind as an *any-to-any* generation model on more challenging cross-modal generation tasks, complemented by our own quantitative comparisons against strong baselines. As shown in Figure 2, FlowBind handles these complex settings while faithfully reflecting the input conditions in its generated outputs.

Interestingly, we observe that fine-grained details from the input (*e.g.*, the Stars and Stripes pattern printed on the table) can reappear in the generated modalities, as shown in the second row of Fig. 2(a). This suggests that FlowBind's learned shared space is expressive enough for cross-modal generation, allowing it to aggregate complementary information from multiple input conditions via latent averaging. We further analyze the effectiveness of this aggregation in Appendix E. Additional qualitative examples are provided in Appendix H and on our project page.

We also conduct quantitative evaluations of FlowBind on many-to-one and one-to-many generation tasks to validate its performance and compare it against baselines. To this end, we construct a synthetic triplet dataset by extending the AudioCaps (Kim et al., 2019) text–audio pairs. Following a protocol similar to OmniFlow, we generate the missing image modality using FLUX.1-schnell

Table 4: Many-to-one generation alignment performances.

| Model | (I+A) → T | | (T+A) → I | | (T+I) → A | |
|---|---|---|---|---|---|---|
| | CLIP (I→T) | CLAP (A→T) | CLIP (T→I) | AIS (A→I) | CLAP (T→A) | AIS (I→A) |
| CoDi | 24.04 | 20.66 | 25.17 | 57.52 | 4.85 | 61.28 |
| OmniFlow | 26.38 | **36.07** | 24.06 | 54.90 | 7.68 | 59.32 |
| **FlowBind** | **27.83** | 35.21 | **25.57** | **57.93** | **28.13** | **76.02** |

Table 5: One-to-many generation alignment performances.

| Model | T → (I+A) | | I → (T+A) | | A → (T+I) | |
|---|---|---|---|---|---|---|
| | CLIP (T→I) | CLAP (T→A) | CLIP (I→T) | AIS (I→A) | CLAP (A→T) | AIS (A→I) |
| CoDi | **26.61** | 10.99 | 25.73 | 58.65 | 18.03 | 57.14 |
| OmniFlow | 24.71 | 12.92 | 26.36 | 63.99 | 36.07 | 54.22 |
| **FlowBind** | 25.02 | **29.12** | **27.98** | **74.34** | **36.79** | **59.99** |

(Black Forest Labs, 2024), conditioned on the text annotations. This yields a triplet (text, image, audio) dataset that enables quantitative evaluation of many-to-many generation.

Tables 4 and 5 report results for the many-to-one and one-to-many settings, respectively, comparing FlowBind against other flow-based models. For each setting, we measure alignment scores for all input–output modality combinations using the same alignment metrics as in the one-to-one evaluation. Note that for all metrics (CLIP, CLAP, AIS), higher values indicate better alignment. Overall, Flow-Bind achieves competitive or superior alignment across tasks. Notably, it exhibits a substantially reduced tendency to ignore specific conditioning modalities in many-to-one settings, demonstrating more balanced cross-modal conditioning. For instance, as shown in Table 4, while CoDi and Omni-Flow tend to disregard the text input during (Text+Image)→Audio generation, FlowBind effectively incorporates both modalities.

## 6 ANALYSIS

**Fixed v.s. Learnable Shared Anchor** The theoretical analysis in Section 3.1 suggests that our training objective induces a meaningful shared latent space that serves as an effective anchor. To further support this claim, we conduct an empirical comparison between having text modality as a fixed anchor and having learnable, shared latent space as an anchor. Similar to the alignment procedure in CoDi, we consider a text-anchoring baseline that directly utilizes text modality as a

Table 6: Comparison of alignment scores between model that uses fixed text anchor and learnable shared anchor. I-A represents the image-audio dataset.

| Model | I → T | A → T | I → A |
|---|---|---|---|
| *Text-anchoring* | 27.94 | 36.72 | 55.48 |
| FlowBind w/o I-A | **30.04** | **37.04** | **61.88** |

fixed anchor. Since the image-audio pair cannot be used in this setting, we compare text-anchoring baseline with a variant of FlowBind that excludes image-audio pair during training. The resulting data-controlled comparison, as reported in Table 6, shows that cross-modal alignment can be improved by introducing learned shared latent space. Specifically, FlowBind variant trained without image-audio pair still outperforms text-anchoring variant in all three measured alignment scores. This suggests that employing learnable shared latent space can be beneficial for cross-modal alignment in general, validating our proposed objective in Eq. 7.

**Analysis on Shared Latent** As mentioned in Section 3.1, our learning objective is designed to produce a shared latent representation that unifies information from all input modalities. We analyze the characteristics of the learned space, hypothesizing that it should exhibit strong cross-modal alignment. To quantitatively assess cross-modal alignment of the shared latent space, we measure the CKNNA metric proposed by Huh et al. (2024), comparing alignment in the shared latent space against that in modality-specific encoded latent spaces. We follow the suggested procedure for computing CKNNA measure, using at most 1024 samples with neighborhood size $k$ set to 10. The analysis is done for text-audio and audio-image alignment, the settings where held-out test set is available.

As shown in Table 7, our representation from the learnable shared latent space exhibits higher alignment scores compared to the latents that are obtained from per-modality encoders. This quantitatively measured improvement in alignment validates our claim that the shared latent space is not merely a co-embedding of features, but rather a truly shared semantic space. Our framework successfully learns a coherent, well-aligned multi-modal latent space that bridges the semantic gap between different modalities, thereby handling complex any-to-any generation tasks effectively.

Table 7: Shared latent space yields higher alignment measured in CKNNA.

| Model | T-A | A-I |
|---|---|---|
| Latent | 0.1965 | 0.1343 |
| Shared Latent | **0.2872** | **0.3026** |

In addition to the quantitative analysis, we conduct a qualitative study of the shared latent space by interpolating between two latents and decoding the latent representations into two different modalities, text and image. As shown in Figure 3, we empirically observe that the shared latent space is indeed a well-aligned, semantically meaningful space, enabling the semantic of decoded image change gradually between two inputs. Additional analysis, including visualizations of the shared latent space, is provided in Appendix F.

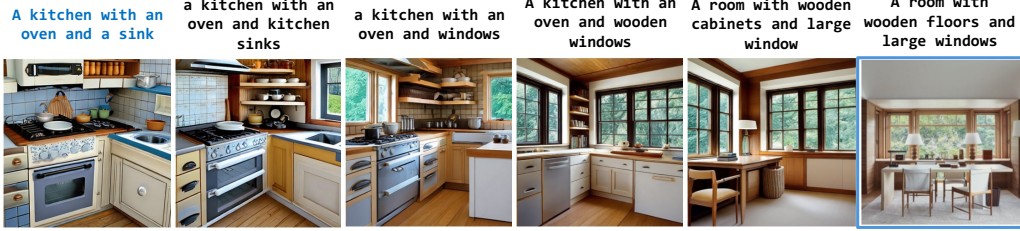

Figure 3: FlowBind's shared latent space learn semantically meaningful space, allowing smooth transition when interpolating between two latents. Data with blue boundary indicates input.

**Extension to Additional Modality** To demonstrate FlowBind's scalability beyond text, images, and audio, we extend it to 3D point clouds and train on Pix3D (Sun et al., 2018). Adding this modality requires only an additonal modality-specific drift network, so the parameter count grows roughly linearly with the number of modalities $N$. Crucially, although training uses only an additional paired image–point clouds dataset, FlowBind not only achieves strong performance on the seen modality combinations (Figure 6), but also generalizes to unseen cross-modal generation tasks, such as text $\rightarrow$ point clouds and point clouds $\rightarrow$ text (Figure 7). This indicates that our central learnable anchor can effectively leverage arbitrarily partially paired data. Detailed settings and qualitative results are provided in Appendix G.

## 7 CONCLUSION

In this work, we introduce a novel framework for any-to-any multi-modal generation that directly addresses the critical limitations of data scarcity and computational complexity inherent in prior methods. By learning from arbitrarily paired data, our model alleviates the need for impractical fully-paired or anchor-based datasets. The core of our approach is a shared latent space trained end-to-end with a single unified flow matching objective. This design not only simplifies the training pipeline but also yields a computationally efficient and highly scalable. Our experiments demonstrate that this approach achieves competitive performance, particularly in non-text-anchored tasks, and learns a well-structured, semantically aligned latent space. Overall, our data-flexible and efficient framework represents a significant step towards building generalist generative models.

**Acknowledgments** This work was in part supported by the National Research Foundation of Korea (RS-2024-00351212 and RS-2024-00436165), the Institute of Information & communications Technology Planning & Evaluation (IITP) (RS-2024-00509279, RS-2022-II220926, and RS-2022-II220959, RS-2019-II190075), and the High-Performance Computing Support Project funded by the Korea government (MSIT). We thank Jaehoon Yoo for thoughtful discussions. We also thank Whie Jung, Dong Hoon Lee, Kiet T. Nguyen, Chanryeol Lee, and Wonjung Kim (all with KAIST) for their help with the qualitative evaluation.

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

## A  PROOFS OF EXPECTED CONDITIONAL VARIANCE

**Setup.** Let $N \in \mathbb{N}$ be the number of modalities and define the shared latent $X := z^* = H_\phi(z^1, \ldots, z^N) \in \mathbb{R}^{d_X}$. Fix $i \in \{1, \ldots, N\}$ and set

$$Y := z^i - z^* \in \mathbb{R}^d, \qquad f(X) := v_{\theta^i}(X, 0) \in \mathbb{R}^d, \qquad m(X) := \mathbb{E}[Y \mid X] = \mathbb{E}[z^i - z^* \mid z^*].$$

Assume square–integrability: $\mathbb{E}\|Y\|_2^2 < \infty$ and $\mathbb{E}\|f(X)\|_2^2 < \infty$. (For vectors, $\mathrm{Var}(Z \mid X) := \mathrm{tr} \, \mathrm{Cov}(Z \mid X)$.)

**Objective at $t=0$.**

$$L_i(\theta, \phi) \;=\; \mathbb{E}\big[\|f(X) - Y\|_2^2\big].$$

**Decomposition with orthogonality.**   Add and subtract $m(X)$ and expand:

$$\|f(X) - Y\|_2^2 = \|f(X) - m(X)\|_2^2 + \|m(X) - Y\|_2^2 + 2\langle f(X) - m(X),\, m(X) - Y\rangle.$$

Taking expectations and conditioning on $X$,

$$\mathbb{E}[\langle f(X) - m(X),\, m(X) - Y\rangle] = \mathbb{E}\big[\big\langle f(X) - m(X),\, \mathbb{E}[m(X) - Y \mid X]\big\rangle\big] = 0,$$

since $\mathbb{E}[m(X) - Y \mid X] = m(X) - \mathbb{E}[Y \mid X] = 0$. Equivalently,

$$m(X) - Y \;\perp\; L^2(\sigma(X)) \quad \text{and} \quad f(X) - m(X) \in L^2(\sigma(X)).$$

Thus,

$$\begin{aligned} L_i(\theta, \phi) &= \mathbb{E}\big[\|Y - m(X)\|_2^2\big] + \mathbb{E}\big[\|f(X) - m(X)\|_2^2\big] \\ &= \underbrace{\mathbb{E}\big[\mathrm{Var}(z^i \mid z^*)\big]}_{\text{unexplained variance}} + \underbrace{\mathbb{E}\big[\|\, v_{\theta^i}(z^*, 0) - \mathbb{E}[z^i - z^* \mid z^*]\,\|_2^2\big]}_{\text{distance to Bayes}}. \end{aligned}$$

Consequently,

$$\min_\theta L_i(\theta, \phi) = \mathbb{E}\big[\mathrm{Var}(z^i \mid z^*)\big], \qquad \text{attained by} \quad v_\theta^{i\,\star}(z^*, 0) = \mathbb{E}[z^i \mid z^*] - z^*.$$

**Summed objective.**   For $S \subseteq \{1, \ldots, N\}$, define

$$L_{t=0}(\theta, \phi) := \mathbb{E}\left[\sum_{i \in S} \|\, v_{\theta^i}(z^*, 0) - (z^i - z^*)\,\|_2^2\right].$$

Summing the above identity over $i \in S$ and using linearity of expectation,

$$L_{t=0}(\theta, \phi) = \sum_{i \in S} \mathbb{E}\big[\mathrm{Var}(z^i \mid z^*)\big] + \sum_{i \in S} \mathbb{E}\big[\|\, v_{\theta^i}(z^*, 0) - \mathbb{E}[z^i - z^* \mid z^*]\,\|_2^2\big],$$

hence

$$\min_\theta L_{t=0}(\theta, \phi) = \sum_{i \in S} \mathbb{E}\big[\mathrm{Var}(z^i \mid z^*)\big], \quad \text{with } v_{\theta^i}{}^\star(z^*, 0) = \mathbb{E}[z^i \mid z^*] - z^* \; \forall i \in S.$$

**Law of Total Variance**   our $t=0$ formulation

$$L_i(\theta, \phi) = \underbrace{\mathbb{E}\big[\mathrm{Var}(z^i \mid z^*)\big]}_{\text{unexplained}} + \underbrace{\mathbb{E}\big[\|\, v_{\theta^i}(z^*, 0) - \mathbb{E}[z^i - z^* \mid z^*]\,\|_2^2\big]}_{\text{distance to Bayes}},$$

there are concrete benefits to reducing it:

**Implication.**   By the Law of Total Variance, $\mathrm{Var}(z^i) = \mathbb{E}[\mathrm{Var}(z^i \mid z^*)] + \mathrm{Var}(\mathbb{E}[z^i \mid z^*])$, so minimizing the unexplained part $\mathbb{E}[\mathrm{Var}(z^i \mid z^*)]$ equivalently maximizes the explained variance $\mathrm{Var}(\mathbb{E}[z^i \mid z^*])$. Because the decomposition holds for any $(\theta, \phi)$, gradients w.r.t. $\phi$ (the encoder) continually act to reduce the summed unexplained variance across modalities.

## B    TRAINING AND INFERENCE

This section presents the detailed training and inference algorithms to provide a clear understanding of each procedural formally.

---

**Algorithm 1:** Training

**Input** : Minibatch $\{\mathbf{z}^{S_b}\}_{b=1}^{B}$;
Aux encoder $H_\phi$;
Flows $\{v_{\theta^i}\}_{i=1}^{N}$ (params $\theta$);
Time sampler $t \sim p(t)$.
**Output:** Loss $\mathcal{L}$

1 **for** *each step* **do**
2     Sample $\{\mathbf{z}^{S_b}\}_{b=1}^{B}$;
3     **for** $b = 1$ **to** $B$ **do**
4        $z_b^* \leftarrow H_\phi(\mathbf{z}^{S_b})$
5     Draw $t_b \sim p(t)$ for $b = 1, \ldots, B$;
6     $\mathcal{L} \leftarrow 0$, $M \leftarrow 0$ **for** $b = 1$ **to** $B$ **do**
7        **for** *each* $i \in S_b$ **do**
8           $z_t \leftarrow (1 - t_b)z_b^* + t_b z_b^i$;
9           $\hat{u} \leftarrow v_{\theta^i}(z_t, t_b)$;
10          $u^\star \leftarrow z_b^i - z_b^*$;
11          $\mathcal{L} \leftarrow \mathcal{L} + \|\hat{u} - u^\star\|_2^2$;
12          $M \leftarrow M + 1$;
13     **if** $M > 0$ **then**
14        $\mathcal{L} \leftarrow \mathcal{L}/M$;
15     **return** $\mathcal{L}$

---

**Algorithm 2:** Inference

**Input** : Sources $S$ with $\{z^i\}_{i \in S}$; target $j$;
Learned flows $\{v_{\theta^i}\}_{i=1}^{N}$; ODESOLVE.
**Output:** $\hat{z}^j$.
```
// Encode sources to shared
   latent  (t : 1→0)
```
1 **for** *each* $i \in S$ **do**
2     $\hat{z}^{*,i} \leftarrow$ ODESOLVE$(z^i, v_{\theta^i}, 1, 0)$;
3 $\hat{z}^* \leftarrow \frac{1}{|S|} \sum_{i \in S} \hat{z}^{*,i}$;
```
// Decode to target  (t : 0→1)
```
4 $\hat{z}^j \leftarrow$ ODESOLVE$(\hat{z}^*, v_{\theta^j}, 0, 1)$;
5 **return** $\hat{z}^j$

---

## C    IMPLEMENTATION DETAILS

### C.1    ENCODERS AND DECODERS FOR EACH MODALITY

For image, we use CLIP (Radford et al., 2021) for visual latent with Stable-UnCLIP (HuggingFace, 2025) as decoder. For audio, we use CLAP (Elizalde et al., 2023) features for conditioning on AudioLDM (Liu et al., 2023). For text, we find that existing text autoencoders such as Optimus (Li et al., 2020) has limited reconstruction abilities. Therefore, we use EmbeddingGemma (Team et al., 2025) for text encoder, and train its decoder with simple reconstruction objective. We use pretrained Gemma3-1B (Team et al., 2025) for initialization and finetune it on two epochs of all texts used in Table 8. Note that these modality-specific encoders and decoders are frozen during the training of FlowBind, thereby not counted as a trained parameters when reporting trainable parameters.

### C.2    ARCHITECTURE

We employ Multi-Layer Perceptron (MLP) for both the flow models $\{v_{\theta^i}\}_{i=1}^{N}$ and the joint estimator $H_\phi$. AdaLN (Peebles & Xie, 2023) is applied to all drift networks to improve time modulation. To ensure dimensional consistency for the flow-based formulation, we set the feature dimensionality to 768 across all modalities. For CLAP-based audio features, we use lightweight two-layer MLP projection and reconstruction modules to map between the CLAP feature space and our shared latent space. For the auxiliary encoder, each modality input is processed by lightweight modality-specific modules, and the resulting outputs are averaged. To enhance training robustness, we incorporate a fixed variance term as a hyperparameter that regularizes the learned representations.

## C.3    Training Dataset

We employ all three types of paired data across text, image and audio. We do not use triplet data in our experiments. We summarize the details about training dataset in Table 8.

Table 8: Training dataset summary.

| Type | Dataset name | Size | Description |
|------|-------------|------|-------------|
| Text–Image | LAION-COCO | 242K | Subset of LAION eV (2025), filtered by aesthetic scores $> 5.0$. Captions are synthetically generated. |
| | Flickr-30k | 30K | Sentence-based image descriptions from Plummer et al. (2015) |
| Text–Audio | AudioCaps v2 | 91K | Natural-language audio captions parsed from YouTube supported by Kim et al. (2019) |
| Audio–Image | VGGSound | 184K | Large-scale audio-visual dataset from YouTube supported by Chen et al. (2020) |

## C.4    Training Recipe

We trained the model for 200K iterations using the Adam optimizer and a global batch size of 1024. The total training process requires approximately 48 GPU-hours on NVIDIA H100. To train each drift network, we normalized the latent representations of each modality to match their respective scales. During training, following Kim et al. (2024), we apply the endpoint (*i.e.*, $t = 1$) velocity prediction objective with probability 0.3, which empirically improves stability.

# D    Evaluation Setup

We evaluate both generation quality and cross-modal alignment using standard, modality-appropriate metrics. Generation quality is measured with FID (Heusel et al., 2017) for images, FAD (Kilgour et al., 2019) for audio, and CIDEr (Vedantam et al., 2015) for text captions. Cross-modal alignment is quantified via pairwise similarity scores: CLIPScore for text–image pairs (Hessel et al., 2021), CLAPScore for text–audio pairs (Elizalde et al., 2023), and Audio-Image Similarity (AIS) for image–audio pairs (Wu et al., 2022). For text–audio and image–audio tasks, we report results on held-out test splits, whereas for text-to-image and image-to-text we follow the widely adopted MS-COCO zero-shot evaluation protocol.

We follow a specialist-baseline protocol, selecting strong off-the-shelf models for each modality pair: SD3-Medium (Esser et al., 2024) (also used as the OmniFlow text-to-image backbone) and FLUX.1 (Black Forest Labs, 2024) for text-to-image; LLaVA-NeXT (Li et al., 2025a) for image-to-text; TangoFlux (Hung et al., 2024) and AudioX (Tian et al., 2025) for text-to-audio; Qwen2-Audio (Chu et al., 2024) for audio-to-text; Seeing&Hearing (Xing et al., 2024) for image-to-audio; and Sound2Vision (Sung-Bin et al., 2024) for audio-to-image.

**Audio-Image-Similarity (AIS)**    We followed SonicDiffusion (Biner et al., 2024) to measure relative AIS on audio-image evaluations. In contrast to other alignment metrics, AIS is a reference-based metric to compensate different scales of measured cosine similarity. For audio-to-image evaluation, AIS is defined as a ratio of audios in testset that achieve worse cosine similarity than conditioning audio, measured with genrated image. For example, AIS is zero if the generated image is not aligned at all and thus shows least cosine similarity among test set audios. The similarity is measured using wav2clip (Wu et al., 2022) audio embedding and image CLIP (Radford et al., 2021) embedding from ViT-B/32 model. We generalize AIS metric for image-to-audio generation in a symmetric way, counting the ratio of images in test set that gives lower cosine similarity compared to conditioning image.

# E    ROBUSTNESS OF PLAIN AVERAGING

In this section, we analyze FlowBind's robustness to competing source modalities in many-to-one generation. We construct a conflict set by randomly pairing audio clips with text prompts that deliberately describe different semantics. We then performed (T + A) → I generation with plain averaging in the shared latent space, and present the results in Figure 4.

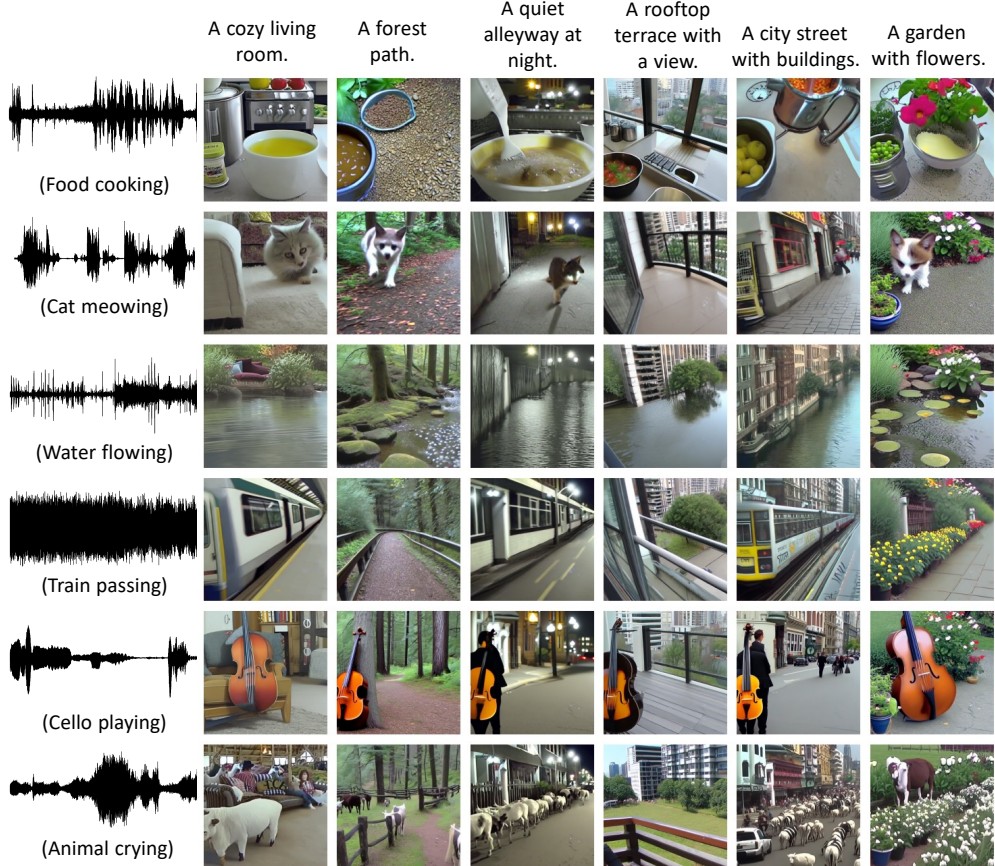

Figure 4: Results on conflicting conditions of {text+audio}-to-image generation.

In this challenging setup, FlowBind faithfully reflects the two conflicting conditions in most cases, rather than collapsing to an incoherent blend or ignoring one modality.

We attribute this robustness to the shared latent space learned by FlowBind. As shown in Table 7, the shared latent exhibits strong cross-modal alignment. We conjecture that structured and semantically coherent geometry of our shared latent space enables even simple averaging to yield stable and meaningful behavior under conflicting conditions.

# F    FLOWBIND LATENT VISUALIZATION

In this section, we further analyze the shared latent space by visualizing the relationship between latent representations and generated content. Specifically, we present a t-SNE visualization of Flow-Bind's shared latent space together with representative generated images.

We sample 5,000 text prompts from the MS-COCO evaluation set, encode each prompt into Flow-Bind's shared latent space, and cluster the resulting latent vectors using $k$-NN with $k=15$ (Figure 5a). For five selected clusters, we decode the five latent vectors closest to each cluster center into images (Figure 5b).

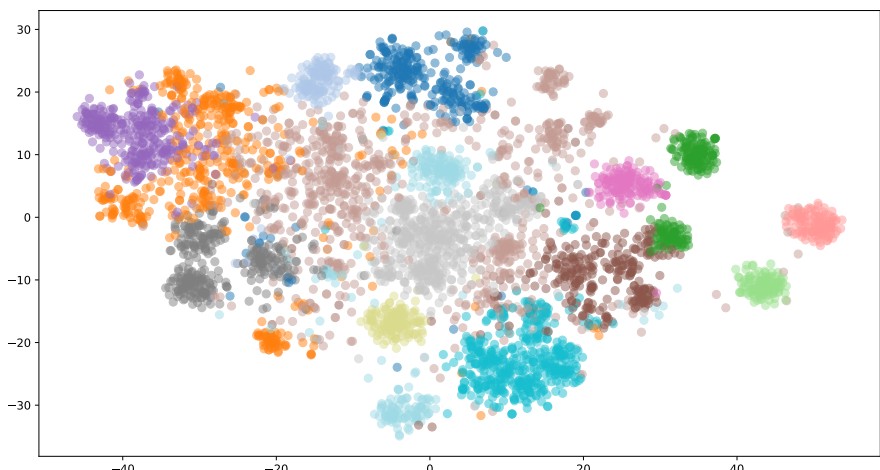

(a) t-SNE visualization of the shared latent space using MS-COCO captions. Clusters are formed by k-NN with $k = 15$.

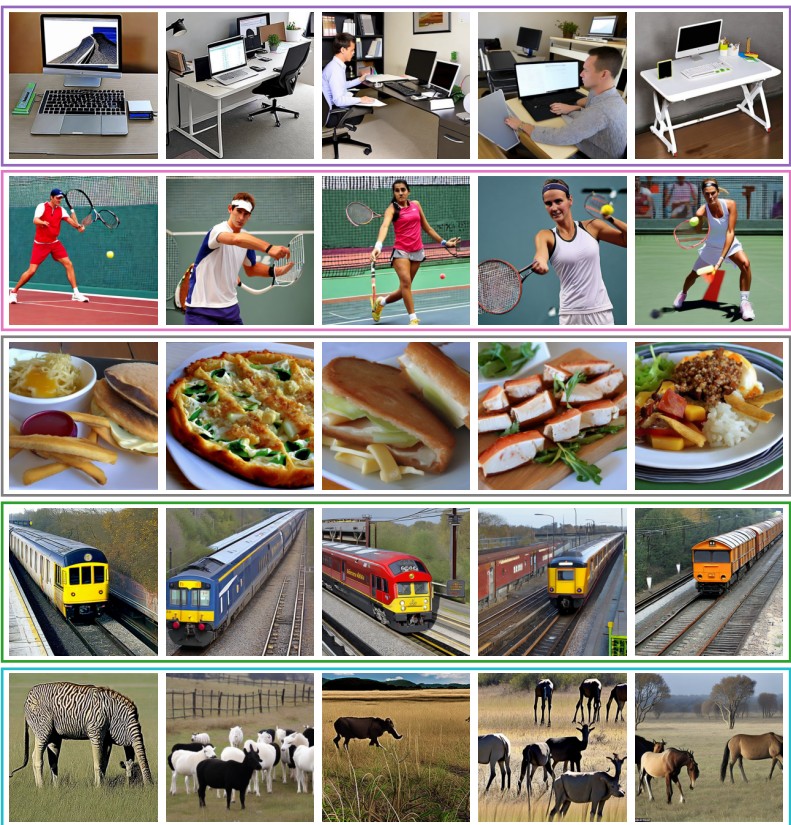

(b) Example image clusters decoded from latent points within a randomly selected cluster. Each color boundary represents a distinct cluster, as shown in 5a.

Figure 5: Visualization of shared latent space of FlowBind and corresponding generated images.

The examples in Figure 5b show that samples drawn from the same cluster in the shared latent space are semantically coherent (e.g., office scenes, tennis players, food dishes, trains, animals), while different clusters capture clearly distinct concepts. These results support that our shared latent space forms representations according to high-level semantics, so that nearby latent points correspond to consistent and meaningful variations in the generated contents.

## G   FLOWBIND WITH ADDITIONAL MODALITY

To demonstrate the scalability of FlowBind, we extend our framework to an additional modality, namely 3D point clouds. We use the Pix3D dataset (Sun et al., 2018), which contains 10k pairs of (Image, Point cloud), and adopt a pre-trained modality-specific autoencoder from (Yang et al., 2019). All other settings are kept the same as in our main experiments (Section 5); adding a new modality only introduces its modality-specific drift network, leading to approximately linear growth in the total number of parameters.

Figure 6 presents the qualitative results for cross-modal generation of image-point clouds, demonstrating strong performance while preserving the geometry of the underlying object and overall consistency of the shape.

More importantly, as shown in Figure 7, FlowBind also achieves reasonable performance on **unseen** cross-modal combinations (e.g., text → point clouds and point clouds → text), indicating that our framework can effectively exploit arbitrarily partially paired data, owing to its central learnable anchor design.

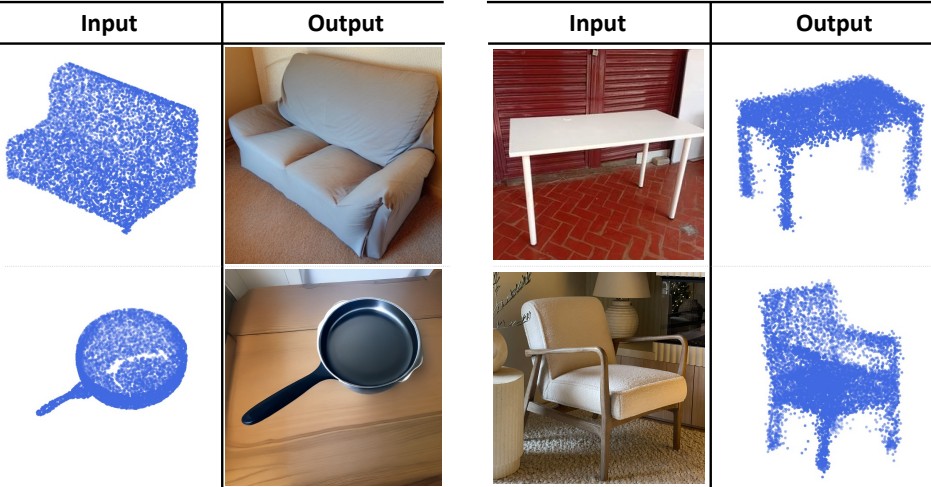

(a) Results on point clouds-to-image generation         (b) Results on image-to-point clouds generation

Figure 6: Cross modal generation results on image–point clouds

| Input | Output |  | Input | Output |
|-------|--------|--|-------|--------|
|       | The chair has pillow in grey fabric. |  | a wooden chair | |
|       | The daylight table with bowls |  | the bookcase with books in it | |

(a) Results on point clouds-to-text generation         (b) Results on text-to-point clouds generation

Figure 7: Cross-modal generation results on text–point clouds. FlowBind handles cross-modal generations *unseen during training* by effectively leveraging arbitrarily partially paired data.

## H  QUALITATIVE RESULTS

In this section, we present more qualitative results on various any-to-any generation, including one-to-one (Figures 8–11), one-to-many (Figures 12–13), and many-to-one (Figures 14–16) generation. Qualitative results on various input and output modality combinations are provided in our project page: `https://yeonwoo378.github.io/official_flowbind`.

It shows that FlowBind faithfully translate input modalities into another modalities while preserving content. Compared to baseline, FlowBind exhibits stronger qualitative results especially on challenging many-to-one generation tasks. Specifically, we observe that the baselines struggle to preserve the heterogeneous contents of different modalities, often failing to produce content of one of two modalities. Compared to this, FlowBind faithfully generates outputs that preserve the content of all input modalities, showcasing the advantage of FlowBind in any-to-any generation tasks.

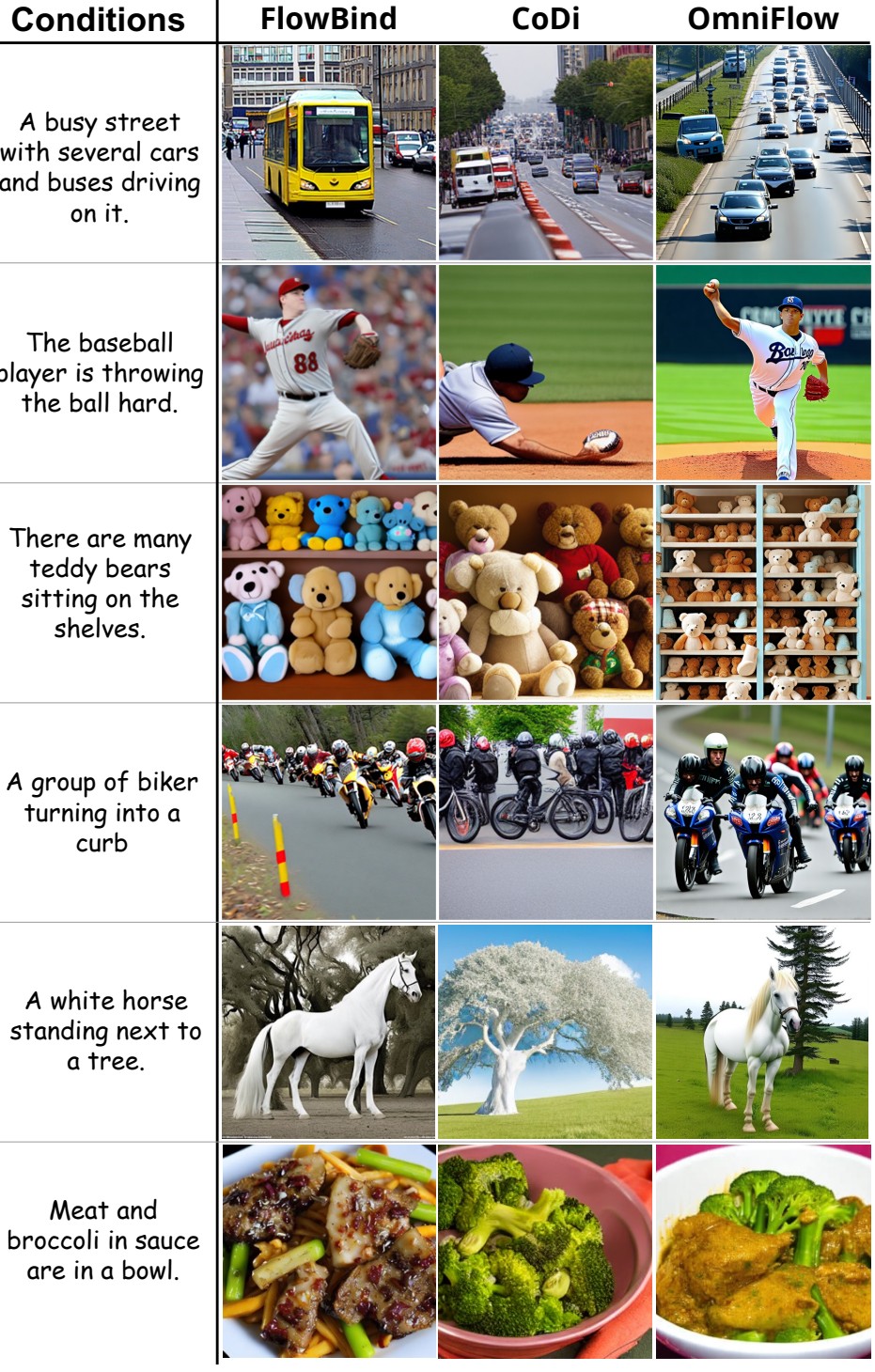

Figure 8: Results on text-to-image generation.

| Conditions | FlowBind | CoDi | OmniFlow |
|---|---|---|---|
|  | A monkey is sitting on top of a rock eating some food. | monkey on a little girl eating a banana on a monkey | a man feeding a baby chimpanzee with a spoon |
|  | People at the beach in the sand are gathered on a sunny day. | three men are beach at the beach home | a group of people and children sitting at a beach with several trucks parked nearby |
|  | A man and woman in formal attire standing next to each other. | couple and dress wear an opera and an attractive man | a man and woman standing together while he is kissing her |
|  | A plate with a pizza and food on top. | pizza and tomato, turkey and pizza | a plate of fires and a piece of chicken on plate |
|  | A bathroom with tile and sink, and a mirror above the wall. | new bathroom chairs get messy so bathroom rooms have bathroom colors | a bathroom with a wall-mounted garden tray |
|  | A young boy is holding a red toy on the beach as an adult looks on. | boy gets his boat while for the swim. | two boys sitting on a surfboard with a yellow kite |

Figure 9: Results on image-to-text generation.

| Conditions | FlowBind | CoDi | OmniFlow |
|---|---|---|---|
| (A man talking as birds are chirping) | Birds vocalize chirping while a man speaks | Man singing for the bird in the tree | A man speaks and a duck quacks |
| (A toilet flushing) | A toilet flushes and water drains | Water reflection person ‹ a bathroom in bathroom› | A cat meows followed by a thump |
| (Humming and vibrating of a tool) | A loud power tool grinding and drilling | Firefighter working on a stick is sharp in the fire | A machine is being used |
| (A person snoring and a man talking) | A person is snoring while sleeping and someone speaks | Two men sleeping and watching the other in the night | A person snoring |
| (A motorboat engine turns on) | a boat engine starting then running | a driver fleeing for a left turn speeding to chase a red horseback from the speeding car | A frog croaks nearby |
| (Light music playing followed by police sirens) | traffic is passing with some distant vehicle sirens going off | a pedestrian walking by a train in a city street while on a city sidewalk walking in the city, where traffic is city. | A siren is going off and helicopters are flying |

Figure 10: Results on audio-to-text generation.

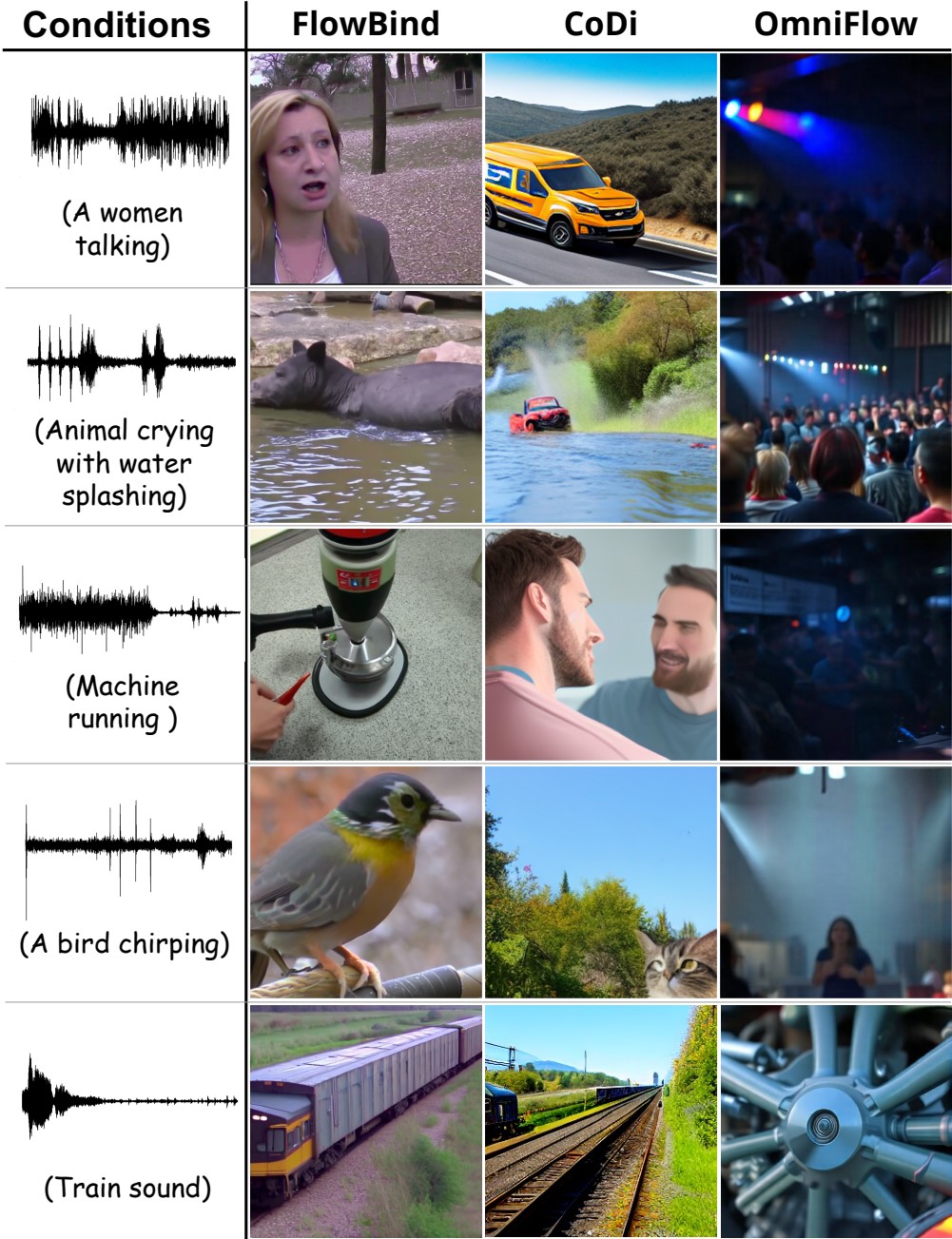

Figure 11: Results on audio-to-image generation.

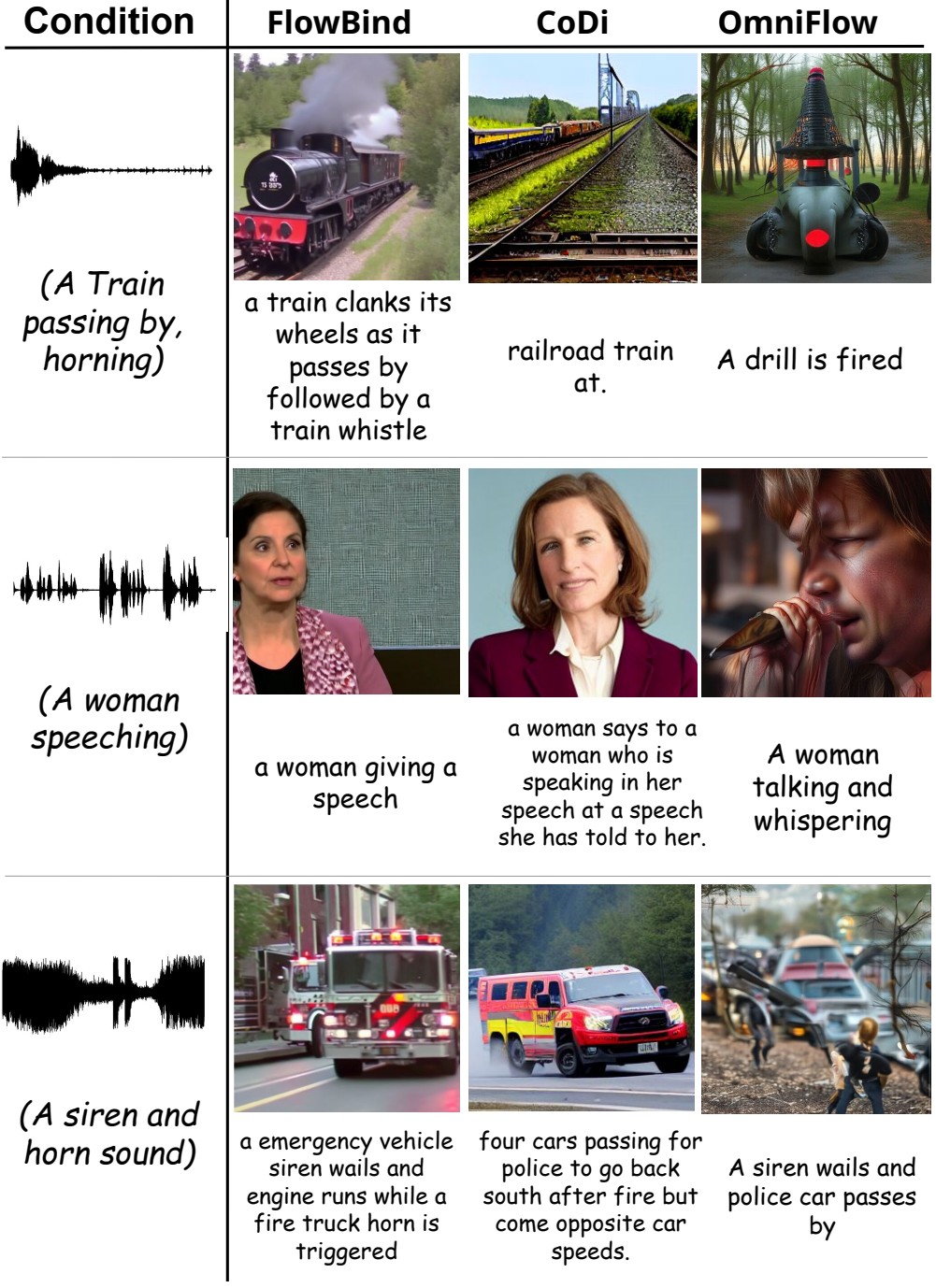

Figure 12: Results on audio-to-{text, image} generation.

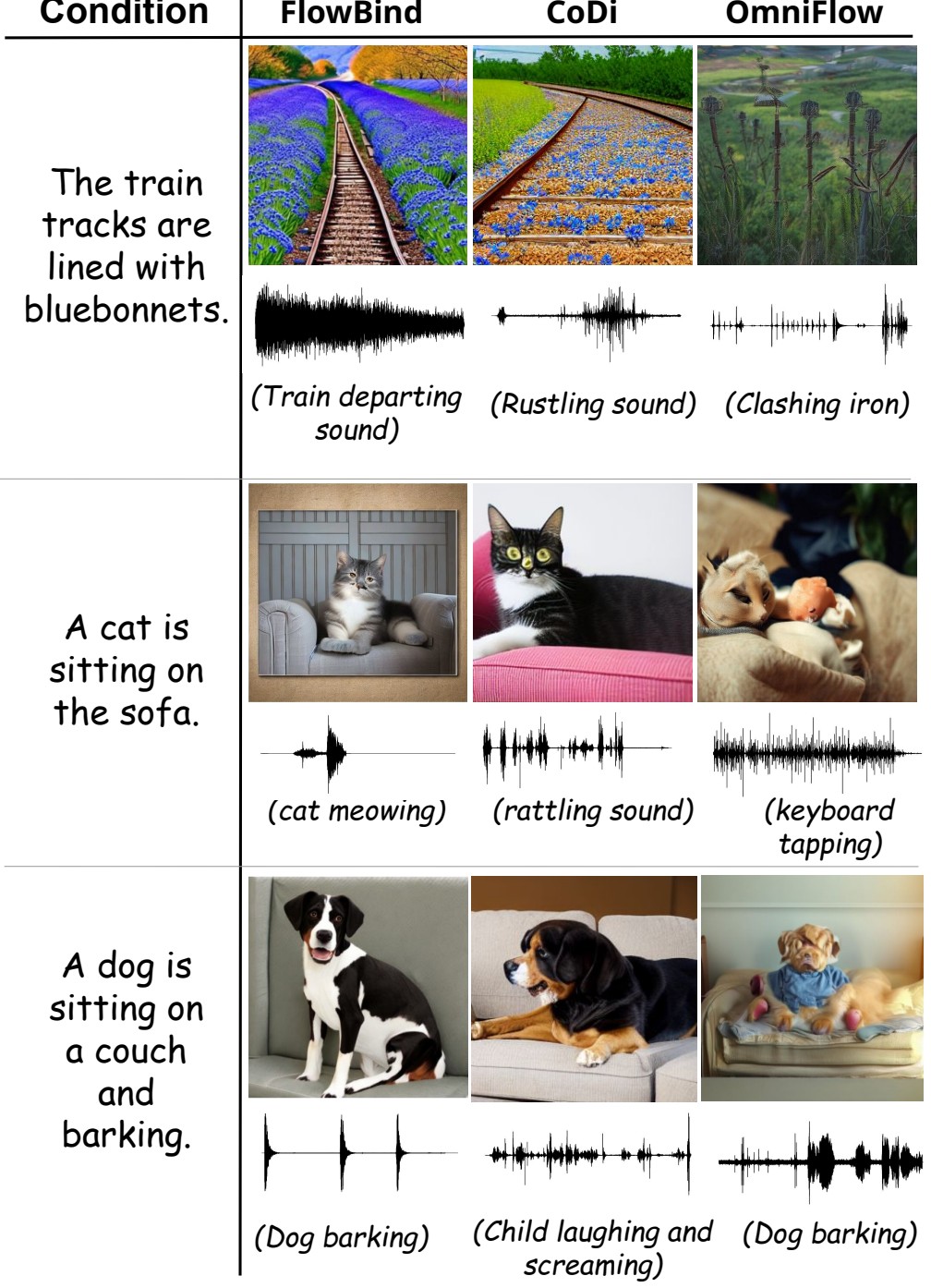

Figure 13: Results on text-to-{image+audio} generation.

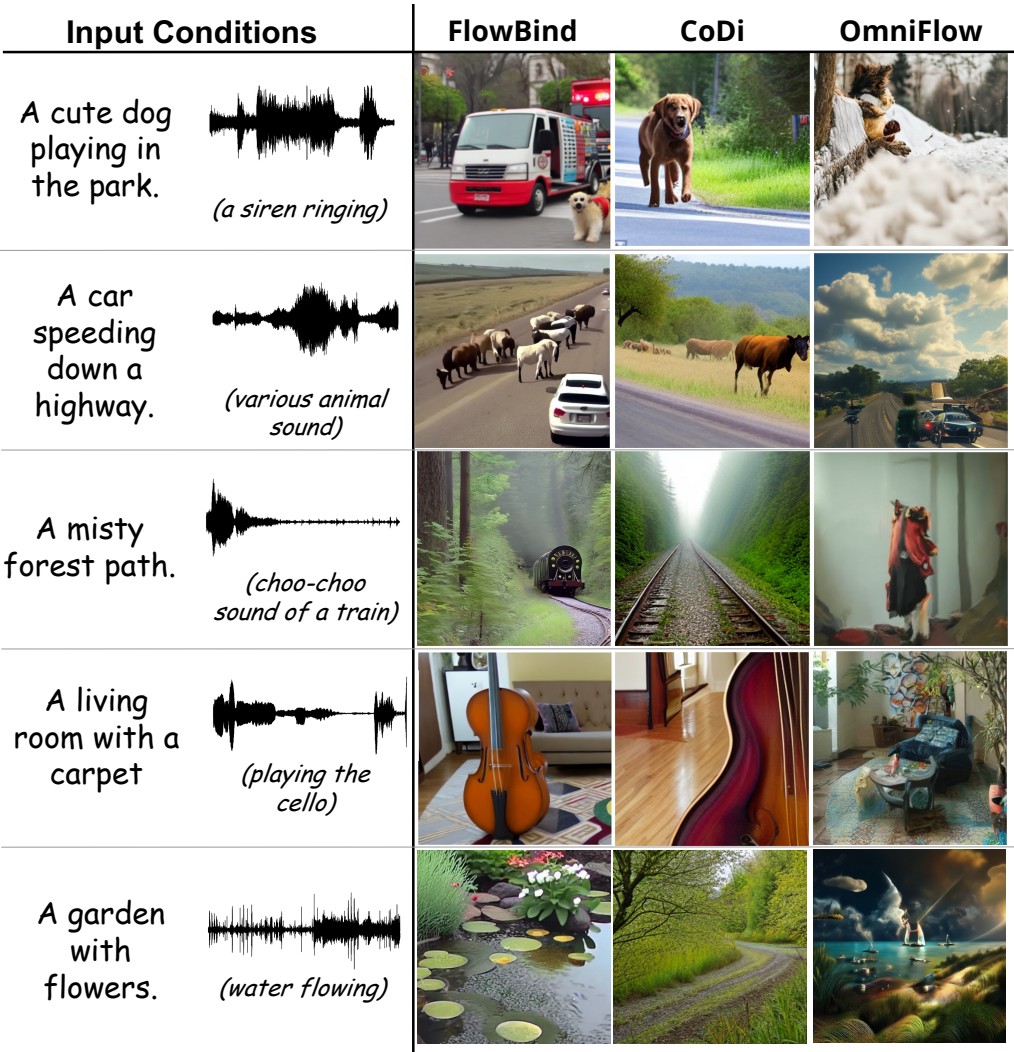

Figure 14: Results on {text+audio}-to-image generation.

| Conditions | | FlowBind | CoDi | OmniFlow |
|---|---|---|---|---|
|  (A bell ringing) | | A bell rings while a motorcycle grooms. | a horseback riding with her woman | A wind blows through the microphone as a horse gallops |
|  (Birds tweeting) | | The birds tweet on the building, after they sing | a beer is pouring wine for a party. | A city street at night with Christmas lights and a fireworks display. |
|  (A vehicle honking) | | An old time photo shows a car horn blows while a man | blue woman reading with a train next. | a black and white painting of a steam locomotive with a man standing next to it |
|  (A man screaming) | | France players celebrating with their team after a victory | soccer players celebrate after they won lisa's goal after their parents left the soccer match on the français after their | A group of boys cheer and scream |
|  (A child talking) | | a child living room with couches and children | boy doesn't appear in the house, that is day. | A child's room with a couch, table and book shelf. |

Figure 15: Results on {image+audio}-to-text generation.

**Input Text:** A sunset over ocean, casting orange and pink hues across the sky.

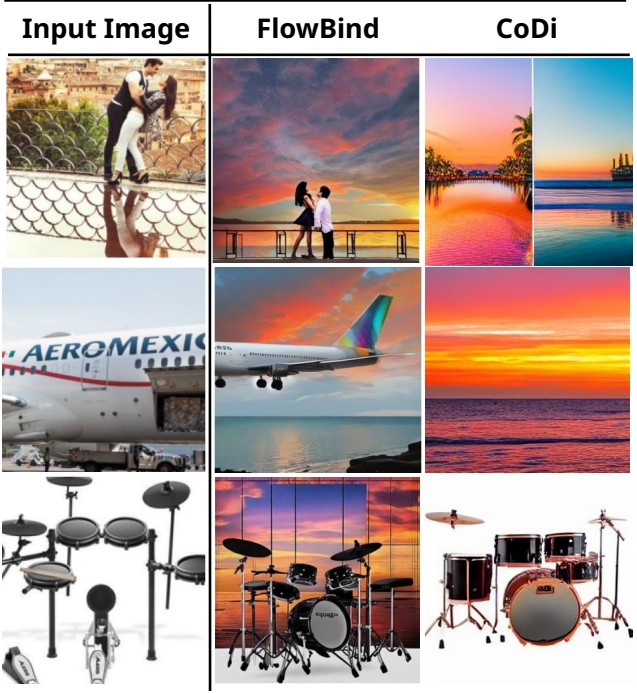

**Input Text:** A quiet garden with colorful flowers blooming.

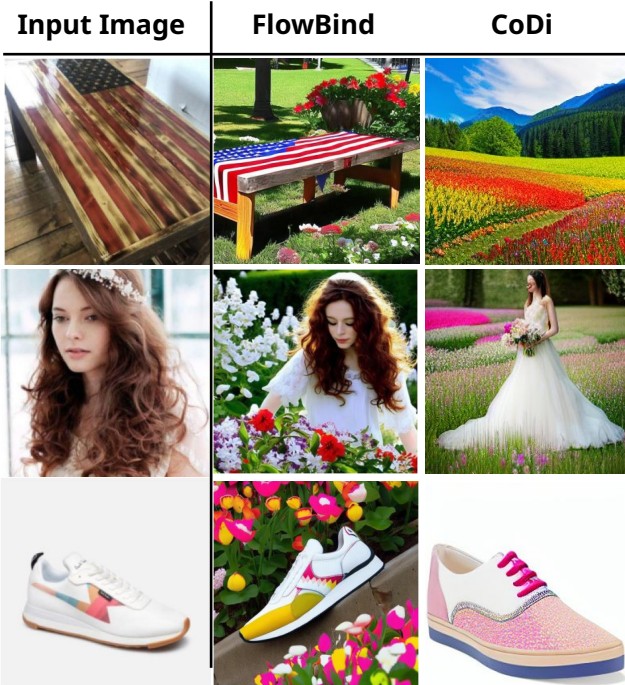

Figure 16: Results on {text+image}-to-image generation.

