# OpenReview forum: "FlowBind: Efficient Any-to-Any Generation with Bidirectional Flows"
_ICLR.cc/2026/Conference — ICLR 2026 Poster_

### Official Review · Reviewer_1rzN · 2025-10-30

**Soundness:** 3
**Presentation:** 4
**Contribution:** 3
**Rating:** 4
**Confidence:** 4

**Summary:**

This paper introduces FlowBind, a novel framework for any-to-any multimodal generation. The core idea is to learn a shared latent space that captures cross-modal semantics, and to connect each modality to this space via a dedicated, invertible flow model. The shared latent and the modality-specific flows—are trained jointly under a single flow-matching objective, which allows the model to learn from arbitrary data pairings (e.g., text-image, image-audio) in a unified, single-stage training process.

The authors claim this design offers significant advantages in efficiency and data flexibility. At inference, the invertible flows function as encoders and decoders, enabling direct translation between any subset of modalities by first mapping the source modality to the shared latent space and then mapping from the latent space to the target modality. The paper presents experiments on text, image, and audio generation tasks, arguing that FlowBind achieves competitive performance against prior methods like CoDi and OmniFlow, while requiring substantially fewer parameters and less training computation.

**Strengths:**

*   **Novel and Elegant Framework Design**: The paper proposes a conceptually clean and novel framework for any-to-any generation. The idea of factorizing the complex joint distribution of multiple modalities through a learnable shared latent space, connected by modality-specific invertible flows, is an elegant and original approach to this problem.

*   **Efficiency and Training Simplicity**: A strength of FlowBind is its computational and data efficiency. By design, the framework avoids the quadratic complexity of joint modeling and simplifies the training process into a single stage with a unified objective. The reported reductions in parameter count and training time are substantial, which is a valuable contribution.

*   **Clarity of Presentation**: The paper is well-written and the core concepts are explained with notable clarity. The illustrations provide an intuitive and effective overview of the training and inference processes, making the proposed methodology easy to follow and understand.

**Weaknesses:**

Despite its novel framework, the paper's experimental evaluation has several weaknesses that make it difficult to accurately assess its claimed "competitive" performance in the context of the current state of the art.

1.  **Outdated and Insufficient Baselines**: The primary weakness of this work is the choice of baselines for comparison, many of which are no longer representative of the state-of-the-art. For instance, in the "Specialists" category of Table 2 and 3, **SD3-Medium** was released over a year ago, and **BLIP2** over two years ago. Similarly, audio models like **AudioLDM-L-full** and **WavCaps** are also established prior work. While comparing to "Generalists" like CoDi is relevant, the rapid progress in the field means that a truly competitive evaluation must include more recent and powerful models. The lack of comparison to any contemporary MLMMs or recent foundation models makes it impossible to gauge how FlowBind's efficiency-focused design trades off against the generation quality achieved by top-tier systems. This omission leaves a significant gap in the evaluation.

2.  **Limited Scope of Modalities and Tasks**: The experiments are confined to only three modalities (text, image, audio) and primarily focus on one-to-one generation tasks.
    *   While the framework is presented as "any-to-any," its true scalability and effectiveness on a larger set of modalities (e.g., video) remain unproven.
    *   The qualitative examples for many-to-one or one-to-many generation (e.g., Figure 2) are interesting but lack quantitative evaluation or strong baseline comparisons, making it difficult to assess performance on these more complex tasks which are central to the "any-to-any" promise.

3.  **Contribution Confounded by Frozen Pre-trained Models**: The framework's true contribution is obscured by its heavy reliance on powerful, frozen pre-trained models such as EmbeddingGemma, CLIP, CLAP, and the Stable-UnCLIP decoder. This design effectively reduces the core learning task to aligning the latent spaces of these pre-existing expert models, rather than learning the full generative process. Consequently, the high generation quality reported may be largely inherited from the strong generative priors of these frozen components, which makes it difficult to isolate and evaluate the novel contribution of the FlowBind framework itself.

**Questions:**

1.  **On the Choice of Baselines**: My primary concern is that the baselines used for comparison are not representative of the current state of the art, making it difficult to assess the true performance of FlowBind. Could you provide a justification for this choice? More importantly, to better situate your results, could you provide comparisons against more recent and powerful models, even on a subset of key metrics? For example:
    *   **Text-to-Image**: How does FlowBind compare to recent diffusion models like **FLUX.1[1]**?
    *   **Image-to-Text**: How does it perform against current vision-language models like **LLaVA-NeXT[2]**?
    *   **Text-to-Audio**: How does it stack up against state-of-the-art audio generators like **Tangoflux[3]**, or **AudioX[4]**?
    *   **Audio-to-Text**: How does its performance compare to captioning systems built on top of strong speech models like **Qwen2-audio**?

2.  **On the Contribution of the FlowBind Framework**: The method relies heavily on very powerful frozen pre-trained models (Gemma, CLIP, Stable-UnCLIP decoder, etc.). This makes it difficult to disentangle the performance gains from your novel flow-based alignment versus the strong priors inherited from these components. To better isolate the contribution of your framework, could you provide an ablation study that uses different encoders/decoders? This would help clarify how much of the final quality is attributable to the learned flow itself.

3.  **On the Scalability of the "Any-to-Any" Claim**: The experiments are limited to three modalities. While the framework is designed to be general, have you performed any preliminary experiments or analysis on how FlowBind would scale to a larger set of modalities, such as video?

[1] Labs B F, Batifol S, Blattmann A, et al. FLUX. 1 Kontext: Flow Matching for In-Context Image Generation and Editing in Latent Space[J]. arXiv preprint arXiv:2506.15742, 2025.

[2] Li F, Zhang R, Zhang H, et al. Llava-next-interleave: Tackling multi-image, video, and 3d in large multimodal models[J]. arXiv preprint arXiv:2407.07895, 2024.

[3] Hung C Y, Majumder N, Kong Z, et al. Tangoflux: Super fast and faithful text to audio generation with flow matching and clap-ranked preference optimization[J]. arXiv preprint arXiv:2412.21037, 2024.

[4] Tian Z, Jin Y, Liu Z, et al. Audiox: Diffusion transformer for anything-to-audio generation[J]. arXiv preprint arXiv:2503.10522, 2025.

---

> ### Author Response · Authors · 2025-11-21
> **Rebuttal Response (1/3)**
>
> We thank the reviewer for the insightful and constructive suggestions on improvements. We address the questions and weaknesses as below.
>
> ### **[W1, Q1] Insufficient Baselines**
>
> We appreciate the reviewer’s concern about the choice of baselines and agree that situating FlowBind with respect to recent powerful models is important. Following the reviewer’s suggestions, we have also evaluated several specialist models for individual cross-modal tasks, The results are summarized in Table A:
>
> **Table A. Fidelity assessment on one-to-one evaluation benchmarks.**
> | Category       | Model          | T → I FID ↓ | I → T CIDEr ↑ | T → A FAD ↓ | A → T CIDEr ↑ | I → A FAD ↓ | A → I FID ↓ |
> |----|------|-------|---|---|------|---|----|
> | **Specialists** | SD3-Medium  | 25.40           | –        | –               | –    | –    | –       |
> |            | FLUX.1  | 22.06           | –                | –               | –                | –               | –               |
> |         | LLaVA-NeXT  | –               | 109.3            | –               | –                | –               | –               |
> |         | TangoFlux   | –               | –                | 1.41            | –                | –               | –               |
> |           | AudioX     | –               | –      | 3.09            | –                | –               | –               |
> |    | Qwen2-Audio| –               | –         | –               | 4.64             | –               | –               |
> |              | Seeing & Hearing| –          | –     | –    | –     | 5.31            | –    |
> |                 |Sound2Vision| –               | –                | –               | –                | –     | 42.55           |
> | **Generalists** | UnifiedIO2-L| 21.54           | 134.7*           | 8.31            | 12.15            | –               | –       |
> |       | CoDi     | 24.80           | 16.40            | 9.84            | 6.62             | 14.58           | 50.4            |
> |     | OmniFlow | 22.97           | 44.20            | 4.20            | 31.79            | 5.67            | 106.03          |
> |         |**FlowBind**   | **17.39**       | **46.26**        | **4.19**        | **55.11**        | **2.50**        | **26.60**       |
>
>
>
>
>
>
> **Table B. Alignment results on one-to-one evaluation benchmarks.**
>
> | Category      | Model            | T$\to$I CLIP ↑ | I$\to$T CLIP ↑ | T$\to$A CLAP ↑ | A$\to$T CLAP ↑ | I$\to$A AIS ↑ | A$\to$I AIS ↑ |
> |--------------|------------------|---------------:|---------------:|---------------:|---------------:|--------------:|--------------:|
> | **Specialists** | SD3-Medium        | 31.60          | –              | –              | –              | –             | –             |
> |              | FLUX.1           | 31.06          | –              | –              | –              | –             | –             |
> |              | LLaVA-NeXT       | –              | 32.14          | –              | –              | –             | –             |
> |              | TangoFlux        | –              | –              | 42.71          | –              | –             | –             |
> |              | AudioX           | –              | –              | 29.29          | –              | –             | –             |
> |              | Qwen2-Audio      | –              | –              | –              | 17.09          | –             | –             |
> |              | Seeing & Hearing | –              | –              | –              | –              | 75.11         | –             |
> |              | Sound2Vision     | –              | –              | –              | –              | –             | 62.39         |
> | **Generalists** | UnifiedIO2-L      | 30.71          | 30.73          | 13.48          | 18.68          | –             | –             |
> |              | CoDi             | 30.26          | 26.24          | 10.79          | 17.94          | 61.55         | 74.26         |
> |              | OmniFlow         | **31.52**      | 27.71          | 24.23          | **45.08**      | 71.71         | 59.22         |
> |              | **FlowBind**     | 28.35          | **29.74**      | **29.08**      | 36.70          | **82.89**     | **78.17**     |
>
>
> For the specialist models in Table 2 and Table 3, some baselines were selected based on prior models used for any-to-any generation. We agree that an update is necessary, and the results have been revised in Table 2 and Table 3 with the inclusion of new specialist models. We retained SD3-Medium to emphasize the strength of the OmniFlow backbone, as discussed in L302 of the paper.
>
> Regarding contemporary multimodal LMs (MLLMs), Table 2 and Table 3 already includes Unified-IO 2 as a recent LLM-based unified model that handles text, image, and audio. Unified-IO 2 exhibits strong performance on text-related tasks. However, it does not support the full set of cross-modal directions that FlowBind targets (e.g., image–audio generation).

---

> ### Author Response · Authors · 2025-11-21
> **Rebuttal Response (2/3)**
>
> ### **[W2, Q2]  Isolating the Contribution of FlowBind's Flow-based Alignment from the Power of Pre-trained Model**
>
> We appreciate the reviewer’s comment. First, we would like to clarify that effectiveness of FlowBind is not simply based on strong generative prior. For example, OmniFlow is initialized with high-capacity pre-trained models such as SD3-Medium, AudioLDM2, and tiny-Llama, each of which is *much stronger* than our encoder/decoder.
>
> The effectiveness of FlowBind primarily arises from two design choices: First, we factorize the flows for each modality through a shared latent space, which allows efficient parameterization, single-stage training, and flexibility to leverage arbitrary paired data (e.g., non-text pair such as Image-Audio data). Second, we employ the semantic features as the latent space of the flow. As outlined in Ln 154-159, it allows the FlowBind to operate in low-dimensional, structured space, enabling lightweight models and data-efficient training. These two choices are both important contributions of our method to achieve highly flexible, compute- and data-efficient training.
>
> To disentangle these two aspects and better show our advantage related to our formulation, we implemented two internal baselines under the same encoder/decoder settings. The first is a joint flow, which conditions modality generation jointly on all other modalities, similar to time-decoupled flow of OmniFlow. The second is a two-stage text-aligned conditional flow, in which all modalities are first aligned to text and the resulting text-aligned representation is fed into the flow as a conditioning vector, similar to CoDi. We train all models using the same training data and report alignment scores on cross-modal generation tasks. The results are presented in Table C below.
>
>
> **Table C. Formulation ablation results on one-to-one generation**
>
> | Method                    | # Param | T$\to$I CLIP | I$\to$T CLIP | T$\to$A CLAP | A$\to$T CLAP | I$\to$A AIS | A$\to$I AIS |
> |---------------------------|---------|--------------|--------------|--------------|--------------|-------------|-------------|
> | **Text-aligned Conditional Flow** | 581M    | 25.43        | 26.12        | 27.91        | 29.77        | 53.64       | 54.72       |
> | **Joint Flow**             | 1.6B    | 26.67        | 28.73        | 23.22        | 31.54        | 80.54       | 76.23       |
> | **FlowBind**               | 568M    | 28.47        | 29.74        | 29.08        | 36.7         | 82.89       | 78.17       |
>
>
>
>
> First, it shows that FlowBind attains clearly better performance than the text-aligned conditioning variant and slightly better overall performance than the full joint model. Compared to the text-aligned conditional flow, FlowBind demonstrates particularly strong performance on non-text settings such as Image→Audio and Audio→Image, although the text-aligned conditional flow is trained with the same amount of image–audio data in the second-stage training. It shows the effectiveness of the FlowBind with the learnable anchor, which flexibly leverages arbitrary, partially paired data in learning both direct alignment and generation in an end-to-end manner. Compared to the joint flow, FlowBind employs much smaller parameters and converges much faster thanks to the factorization. Yet, it attains slightly stronger performance despite the factorization, presumably because the factorization with the learned anchor provides better generalization performance especially under data-efficient settings.
>
> We appreciate the reviewer’s comment and believe that the additional results are helpful to understand the benefits of the FlowBind framework. We will incorporate the results in the final version of the main paper.

---

> ### Author Response · Authors · 2025-11-21
> **Rebuttal Response (3/3)**
>
> ### **[W2-2] Lacking quantitative evaluation on one-to-many and many-to-one experiments.**
>
> FlowBind is precisely designed to support one-to-one, one-to-many, and many-to-one generation, rather than focusing only on single cross-modal generation. We acknowledge that Table 2 and Table 3 mainly report one-to-one generation results, which may give the impression that we focus on single cross-modal generation; this is largely due to the lack of established many-to-many quantitative benchmarks. To better reflect the one-to-many and many-to-one generation capability, we additionally conduct a quantitative evaluation by constructing a synthetic triplet dataset: starting from AudioCaps text–audio pairs and generating the missing image modality with FLUX.1-schnell conditioned on the text. This yields a (text, audio, image) triplet dataset on which we report many-to-one and one-to-many alignment scores in Table D and E, showing that FlowBind handles multi-modal inputs and outputs competitively.
>
> **Table D. Many-to-one generation alignment performances (Appendix Table 7)**
>
> | Method | (T+A) $\to$ I | | (T+I) $\to$ A | | (I+A) $\to$ T | |
> |----------|---------------|---------------|---------------|---------------|---------------|---------------|
>  | | CLIP (T$\to$I) | AIS (A$\to$I) | CLAP (T$\to$A) | AIS (I$\to$A) | CLIP (I$\to$T) | CLAP (A$\to$T) |
> | CoDi | 25.17 | 57.52 | 4.85 | 61.28 | 24.04 | 20.66 |
> | OmniFlow | 24.06 | 54.90 | 7.68 | 59.32 | 26.38 | 36.07 |
> | **FlowBind** | 25.57 | 57.93 | 28.13 | 76.02 | 27.83 | 35.21 |
>
>  **Table E. One-to-many generation alignment performances (Appendix Table 8)**
> | Method | T$\to$(I + A) | | I$\to$(T + A) | | A$\to$(T + I) | |
> |-----------|---------------|---------------|---------------|---------------|---------------|---------------|
> | | CLIP (T$\to$I) | CLAP (T$\to$A) | CLIP (I$\to$T) | AIS (I$\to$A) | CLAP (A$\to$T) | AIS (A$\to$I) |
> | CoDi | **26.61** | 10.99 | 25.73 | 58.65 | 18.03 | 57.14 |
> | OmniFlow | 24.71 | 12.92 | 26.36 | 63.99 | 36.07 | 54.22 |
> | **FlowBind** | 25.02 | **29.12** | **27.98** | **74.34** | **36.79** | **59.99** |
>
>
>
> ### **[W2-1, Q3] Results on more modality**
> We appreciate the reviewer’s suggestion. While we followed the settings from OmniFlow to choose image, audio, and text to demonstrate any-to-any generation, we agree that it would be beneficial to demonstrate our method in more modalities. We will incorporate additional modalities in our experiment and report the results within the remaining rebuttal period.

---

> ### Author Response · Authors · 2025-11-28
> **Rebuttal Response with Additional Experiments**
>
> ### **[W2, Q3] Scalability of FlowBind to additional modality**
>
> We appreciate the reviewer’s insightful comment. To demonstrate the scalability of FlowBind, we extend our framework to an additional modality, namely 3D point clouds. We use the Pix3D dataset [1], which contains 10k pairs of (Image, Point cloud), and adopt a pre-trained modality-specific autoencoder from [2]. All other settings follow our main experiments; adding a new modality only introduces its modality-specific drift network, leading to approximately linear growth in the total number of parameters.
>
> The qualitative results for Image $\leftrightarrow$ Point cloud are reported in Appendix J. Figure 14 presents the generation results of image-point clouds cross-modal generation, demonstrating strong performance while preserving the underlying object geometry and overall shape consistency. More importantly, as shown in Figure 15, FlowBind also achieves reasonable performance on \textit{unseen} cross-modal combinations (e.g., Text $\to$ Point cloud and  Point cloud $\to$ Text), indicating that our framework effectively leverages arbitrarily partially paired data.
>
> Overall, FlowBind scales to additional modalities and can be trained effectively from arbitrarily partially paired data. This is enabled by the shared latent anchor design of our framework, which leads to symmetric handling of all modalities and thereby supports its overall scalability.
>
> [1] Pix3d: Dataset and methods for single-image 3d shape modeling (CVPR 2018)
>
> [2] Pointflow: 3d point cloud generation with continuous normalizing flows (ICCV 2019)

---

### Official Review · Reviewer_XtMR · 2025-10-31

**Soundness:** 4
**Presentation:** 4
**Contribution:** 3
**Rating:** 8
**Confidence:** 3

**Summary:**

This paper introduces FlowBind, a flow-based generative model designed for efficient any-to-any multimodal generation. The core innovation lies in the introduction of a learnable shared latent space that captures cross-modal commonalities, enabling each modality to connect to this latent via its own invertible flow. The model is trained end-to-end under a single flow-matching objective, eliminating the need for multi-stage pipelines or fully-paired datasets. Experiments across text, image, and audio modalities demonstrate that FlowBind achieves competitive performance with significantly reduced computational and data requirements compared to existing baselines like CoDi and OmniFlow.
The contribution can be summarized as:
1. FlowBind enables any-to-any generation by factorizing multimodal interactions through a shared latent space, reducing computational complexity and supporting training with partially paired data.
2. The model jointly optimizes the shared latent and modality-specific flows under a single objective, avoiding complex multi-stage training pipelines.
3. FlowBind achieves strong performance with only 568M trainable parameters, 48 GPU-hours of training, and a fraction of the data required by prior methods.

**Strengths:**

1. The paper is well-structured, clearly written, and easy to understand. The idea of a learnable shared latent as a dynamic anchor for multimodal flows is both solid and well-motivated, addressing limitations of fixed anchors and joint conditioning models.
2. The evaluation covers six one-to-one cross-modal tasks (text-image, text-audio, image-audio, and their reverses) and complex many-to-many generation, with rigorous metrics (FID, FAD, CIDEr, CLIP, CLAP, AIS) for both quality and alignment. The qualitative results (e.g., preserving fine-grained details in multi-modal inputs) effectively showcase the framework’s expressiveness.
3. The empirical results clearly demonstrate FlowBind’s advantages: requiring up to 6× fewer parameters, training 10× faster, and using less than 10% of the data while achieving competitive or better generation quality. The single-stage training pipeline (avoiding multi-stage alignment/generation decoupling) further enhances its practicality for real-world scenarios.

**Weaknesses:**

1. While the paper empirically demonstrates that FlowBind outperforms larger models trained on more data, it lacks a thorough and detailed analysis explaining why this is the case. The authors should provide a deeper investigation into the factors contributing to this unreasonable efficiency.
2. Relying on CLIP (and analogous pre-trained encoders) introduces inherent limitations in capturing fine-grained details, directly constraining generation accuracy. CLIP’s training is driven by coarse-grained semantic alignment between images and text. This may lead to systematic loss of fine visual patterns—such as object positions, subtle textures, or local structure variations.
3. Although the paper includes a basic interpolation analysis (Figure 3) and alignment metrics (Table 5), it lacks a comprehensive interpretability analysis of the learned shared latent space. The authors should provide a visualization of the relationship between the latents and generated contents.
4. (Not Important) The experiments are limited to text, image, and audio. The paper does not address how FlowBind would scale to additional modalities. Questions remain about potential bottlenecks in the shared latent space or computational cost as the number of modalities grows.

**Questions:**

see Weaknesses

---

> ### Author Response · Authors · 2025-11-21
> **Rebuttal Response (1/2)**
>
> We thank the reviewer for the insightful and constructive suggestions on improvements. We address the questions and weaknesses as below.
>
> ### **[W1] Factors Contributing to FlowBind's Efficiency and Performance Compared to Larger Baselines**
>
> We appreciate the reviewer’s request for a clearer explanation of why FlowBind achieves strong performance with comparatively low cost. The efficiency of FlowBind primarily arises from two design choices: First, we factorize the flows for each modality through a shared latent space, which allows efficient parameterization, single-stage training, and flexibility to leverage arbitrary paired data (e.g., non-text pair such as Image-Audio data). Second, we employ the semantic features as the latent space of the flow. As outlined in L154-159, it allows the FlowBind to operate in low-dimensional, structured space, enabling lightweight models and data-efficient training. Note that our efficiency is not simply based on strong generative prior: for instance, OmniFlow employs high-capacity pre-trained models such as SD3-Medium, AudioLDM2, and tiny-Llama, each of which is much stronger than our encoder/decoder.
>
> To disentangle these two aspects and better show our advantage related to our formulation, we implemented two internal baselines under the same encoder/decoder settings. The first is a joint flow, which conditions modality generation jointly on all other modalities, similar to time-decoupled flow of OmniFlow. The second is a two-stage text-aligned conditional flow, in which all modalities are first aligned to text and the resulting text-aligned representation is fed into the flow as a conditioning vector, similar to CoDi. We train all models using the same training data and report alignment scores on cross-modal generation tasks. The results are presented in Table A below.
>
>
> **Table A. (One-to-one) Formulation Ablation**
>
> | Method     | # Param | T$\to$I CLIP | I$\to$T CLIP | T$\to$A CLAP | A$\to$T CLAP | I$\to$A AIS | A$\to$I AIS |
> |---|----|-----|----|---------|---|-----|-----|
> | **Text-aligned Conditional Flow** | 581M    | 25.43        | 26.12        | 27.91        | 29.77  | 53.64       | 54.72       |
> | **Joint Flow** | 1.6B    | 26.67        | 28.73        | 23.22        | 31.54        | 80.54       | 76.23       |
> | **FlowBind**  | 568M    | 28.47        | 29.74        | 29.08        | 36.70        | 82.89       | 78.17       |
>
>
>
> First, it shows that FlowBind attains clearly better performance than the text-aligned conditioning variant and slightly better overall performance than the full joint model. Compared to the text-aligned conditional flow, FlowBind demonstrates particularly strong performance on non-text settings such as Image→Audio and Audio→Image, although the text-aligned conditional flow is trained with the same amount of image–audio data in the second-stage training. It shows the effectiveness of the FlowBind with the learnable anchor, which flexibly leverages arbitrary, partially paired data in learning both alignment and generation in an end-to-end manner. Compared to the joint flow, FlowBind employs much smaller parameters and converges much faster thanks to the factorization. Yet, it attains stronger performance despite the factorization, presumably because the factorization with the learned anchor provides better generalization performance especially under data-efficient settings.
>
> We appreciate the reviewer’s comment and believe that the additional results are helpful to understand the benefits of the FlowBind framework. We will incorporate the results in the final version of the main paper.
>
> ### **[W2] Limitations of CLIP and Pre-trained Encoders in Capturing Fine-Grained Details**
>
> We appreciate the reviewer’s insight and agree that CLIP, primarily trained for coarse semantic alignment, may under-emphasize fine-grained visual details. However, we would like to emphasize that one of our contributions is the novel formulation of any-to-any generation that introduces factorized flows on top of learnable shared latent space. While FlowBind is built upon pretrained encoders, it is not tailored for a specific choice of encoder. We believe analyzing the effectiveness of FlowBind under various configurations to be an interesting direction for future work.

---

> ### Author Response · Authors · 2025-11-21
> **Rebuttal Response (2/2)**
>
> ### **[W3] Visualization of the Learned Shared Latent Space and Its Relationship with Generated Content**
>
> To further analyze the interpretability of the shared latent space and visualize the relationship between latents and generated content, we provide an additional t-SNE plot of FlowBind’s shared latent space along with representative generated images. The results are presented in Appendix I.
> In detail, we sampled 5,000 random text prompts from the MS-COCO evaluation set, encoded each prompt into FlowBind’s shared latent space, and then performed clustering in this space using k-NN with k=15 (Figure 13(a)). For some random clusters, we decoded the top 5 center features into images (Figure 13(b)). The images within the same cluster appear semantically very similar, indicating that the shared latent space aligns meaningful semantic structure across modalities and that nearby latents correspond to coherent variations in the generated content.
> If the reviewer would like to see a different setup or additional results beyond our current experiments, we would be happy to provide them in the remaining rebuttal period.
>
> ### **[W4] Scaling FlowBind to Additional Modalities**
>
> We appreciate the reviewer’s suggestion. While we followed the settings from OmniFlow to choose image, audio, and text to demonstrate any-to-any generation, we agree that it would be beneficial to demonstrate our method in more modalities. We will incorporate additional modalities in our experiment and report the results within the remaining rebuttal period.

---

> ### Author Response · Authors · 2025-11-28
> **Rebuttal Response with Additional Experiments**
>
> ### **[W4] FlowBind with additional modality**
>
> We appreciate the reviewer’s insightful comment. To demonstrate the scalability of FlowBind, we extend our framework to an additional modality, namely 3D point clouds. We use the Pix3D dataset [1], which contains 10k pairs of (Image, Point cloud), and adopt a pre-trained modality-specific autoencoder from [2]. All other settings follow our main experiments; adding a new modality only introduces its modality-specific drift network, leading to approximately linear growth in the total number of parameters.
>
> The qualitative results for Image $\leftrightarrow$ Point cloud are reported in Appendix J. Figure 14 presents the generation results of image-point clouds cross-modal generation, demonstrating strong performance while preserving the underlying object geometry and overall shape consistency. More importantly, as shown in Figure 15, FlowBind also achieves reasonable performance on \textit{unseen} cross-modal combinations (e.g., Text $\to$ Point cloud and  Point cloud $\to$ Text), indicating that our framework effectively leverages arbitrarily partially paired data.
>
> In summary, FlowBind can be extended to additional modalities with approximately linear growth in the number of parameters. The qualitative results further confirm that FlowBind learns cross-modal bridging effectively, implying that our learnable shared latent space successfully integrates information from all four modalities.
>
> [1] Pix3d: Dataset and methods for single-image 3d shape modeling (CVPR 2018)
>
> [2] Pointflow: 3d point cloud generation with continuous normalizing flows (ICCV 2019)

---

### Official Review · Reviewer_Hcpe · 2025-11-01

**Soundness:** 3
**Presentation:** 3
**Contribution:** 3
**Rating:** 6
**Confidence:** 5

**Summary:**

This work replaces the usual logic of any-to-any generation, “everything must talk to text” or “one giant joint flow over all modalities”, with a learnable shared latent and one invertible flow per modality, all trained with one flow-matching loss. On text–image–audio cross-modal generation, they beat/are competitive with CoDi/OmniFlow on 6 one-to-one tasks, especially strong on image-audio, while using 6× fewer params and ~10× less compute than OmniFlow (568M, 48 GPU-hr on H100).

**Strengths:**

1. One shared latent + per-modality flows = no quadratic blow-up over modalities, unlike joint-time rectified flows.
2. No “align to text first, then joint train” and also no post-merge stage. This offers engineering advantage.
3. Each flow only needs its modality + latent, so they can actually use image–audio pair easily.
4. Simple averaging in latent still gives multi-condition outputs

**Weaknesses:**

1. For competing source modalities (conflicting audio + image like two contradicting semantics), how robust is plain averaging?
2. This work focuses more on single cross-modal generation not joint-outputs generation.
3. ODE cost at inference is per modality. There’s still ODE solves on both sides. And there is not enough report on runtime / steps / efficiency as the main claim of this framework is efficiency.

**Questions:**

Right now you average the per-modality latents. Did you try learned fusion (attention / confidence weighting) and does it improve conflicting-condition cases (e.g. image says “dog” but audio says “car”)?

---

> ### Author Response · Authors · 2025-11-21
> **Rebuttal Response**
>
> We thank the reviewer for the insightful and constructive suggestions on improvements. We address the questions and weaknesses as below.
>
> ### **[W1, Q1] Robustness of Plain Averaging / Exploring Averaging and Learned Fusion for Per-Modality Latents**
>
> We appreciate the comment. First, we kindly remind the reviewer that Figure 9 and 10 in the appendix demonstrate the results with two competing source modalities. Specifically, to evaluate how well FlowBind preserves semantics in source modalities, we intentionally constructed two sources with conflicting semantics. We observed that FlowBind faithfully preserves semantics of both source modalities, showing the robustness of simple averaging.
>
> To further validate the robustness of competing source modalities, we constructed a conflict set by randomly pairing audio clips with text deliberately describing different semantics. We then performed (T + A) → I generation with plain averaging in the shared latent space, and present the results in Figure 12 in Appendix H. In this challenging setup, FlowBind faithfully reflects the two conflicting conditions in most cases, rather than collapsing to an incoherent blend or ignoring one modality.
>
> We believe this robustness is the benefits of the shared latent space learned by FlowBind: as mentioned in Table 5, the shared latent already achieves strong cross-modal alignment. Because the shared latent space is well-structured and semantically aligned, even simple averaging of per-modality leads to stable and meaningful behavior under conflicting conditions.
>
>
>
> ### **[W2] This work focuses more on single cross-modal generation not joint-outputs generation.**
>
> Our goal with FlowBind is precisely to build a generalist model that supports one-to-one, one-to-many, and many-to-one generation, rather than focusing only on single cross-modal generation. We acknowledge that Table 2 and 3 mainly report one-to-one results, which may give the impression that we emphasize single cross-modal generation tasks. This is due to the lack of established many-to-many quantitative benchmarks.
> To better reflect the joint-output capability, we have additionally conducted a new quantitative many-to-many evaluation. Concretely, we construct a synthetic triplet dataset by starting from the AudioCaps text–audio pairs and generating the missing image modality using FLUX.1-schnell conditioned on the text annotations. This yields a (text, audio, image) triplet dataset that enables quantitative evaluation of many-to-many generation, and we report many-to-one and one-to-many alignment scores in the rebuttal (Tables A and B). These results demonstrate that FlowBind can handle joint multi-modal inputs and outputs in a principled and competitive way, beyond the single cross-modal generation tasks. We appreciate the reviewer’s comment and will include the results in the main paper.
>
>
> **Table A. Many-to-one generation alignment performances (Appendix Table 7)**
> | Method | (T+A) $\to$ I | | (T+I) $\to$ A | | (I+A) $\to$ T | |
> |-----|-------|----|-----|----|---|-|
>  | | CLIP (T$\to$I) | AIS (A$\to$I) | CLAP (T$\to$A) | AIS (I$\to$A) | CLIP (I$\to$T) | CLAP (A$\to$T) |
> | **CoDi** | 25.17 | 57.52 | 4.85 | 61.28 | 24.04 | 20.66 |
> | **OmniFlow** | 24.06 | 54.90 | 7.68 | 59.32 | 26.38 | **36.07** |
> | **FlowBind** | **25.57** | **57.93** | **28.13** | **76.02** | **27.83** | 35.21 |
>
>  **Table B. One-to-many generation alignment performances (Appendix Table 8)**
> | Method | T$\to$(I + A) | | I$\to$(T + A) | | A$\to$(T + I) | |
> |--|---|----|-|---|---|----|
> | | CLIP (T$\to$I) | CLAP (T$\to$A) | CLIP (I$\to$T) | AIS (I$\to$A) | CLAP (A$\to$T) | AIS (A$\to$I) |
> | **CoDi** | **26.61** | 10.99 | 25.73 | 58.65 | 18.03 | 57.14 |
> | **OmniFlow** | 24.71 | 12.92 | 26.36 | 63.99 | 36.07 | 54.22 |
> | **FlowBind** | 25.02 | **29.12** | **27.98** | **74.34** | **36.79** | **59.99** |
>
>
>
> ### **[W3]  Comparison of Inference Time and Computational Cost**
> We agree that reporting the actual inference cost is important. In our current setup, FlowBind’s ODE cost is moderate in practice, partly because the shared latent has a relatively small dimensionality. Below, in Table C, we report the wall-clock runtime and the number of function evaluations (NFEs) for a single text-to-image generation, measured on the same environment (NVIDIA RTX A6000 GPU), and compare FlowBind against CoDi and OmniFlow. Note that, in the image and audio generation setting, FlowBind uses 20 ODE steps for each modality decoder.
>
>
> **Table C. Inference cost comparison**
>
> | Method     | Runtime | NFEs            |
> |------------|---------|-|
> | **CoDi**       | 3.86s   | 50 DDIM         |
> | **OmniFlow**   | 7.61s   | 50 Euler        |
> | **FlowBind**   | 1.62s   | 10+10 Euler (+20 DPM) |

---

> > ### Comment · Reviewer_Hcpe · 2025-11-25
> >
> > Thanks the authors for answering my concerns regarding plain average and clarifying the appendix source. The joint generation results look good. The inference cost seems pretty promising. I have no other concerns.

---

> > > ### Author Response · Authors · 2025-11-27
> > >
> > > We sincerely appreciate the reviewer’s constructive feedback and the positive adjustment of the score.

---

### Official Review · Reviewer_kcBk · 2025-11-02

**Soundness:** 2
**Presentation:** 2
**Contribution:** 2
**Rating:** 6
**Confidence:** 3

**Summary:**

The paper proposes FlowBind, a flow‑based any‑to‑any generative framework that replaces a fixed Gaussian prior with a learnable, shared latent “anchor.” Each modality connects to this anchor via its own invertible drift network; all drifts and the auxiliary encoder producing the shared latent are trained jointly with a single flow‑matching objective. At inference, per‑modality ODEs are integrated backward to the shared latent and forward to the target modality, enabling many‑to‑many translation. The implementation freezes modality encoders/decoders (EmbeddingGemma for text, CLIP+Stable‑UnCLIP for images, CLAP for audio) and uses small MLP drifts.

**Strengths:**

- [S1] The idea is simple and easy to follow.
- [S2] Promising efficiency: strong results with a lightweight model, fewer GPU‑hours than previous works.
- [S3] Competitive results on several quantitative measurements (e.g., lower FID for T→I and A→I than baselines).

**Weaknesses:**

- [W1] Insufficient ablations for core claims. The main “analysis” section contains only two small studies: fixed text‑anchor vs learnable shared anchor (Table 4) and a CKNNA alignment probe (Table 5). There are no ablations on (i) the shared‑latent aggregation rule (simple averaging vs alternatives such as weighted averaging, or learned averaging), (ii) latent dimensionality or encoder/decoder choices, or (iii) the fraction of partially paired data. Without these, it is hard to attribute gains to the proposed factorization rather than to frozen backbones or evaluation choices.
- [W2] Partial‑pairing claim not quantified. The paper argues for learning from arbitrary partially paired data, but does not vary the paired/unpaired ratios or show robustness curves. This leaves the “data‑flexible” claim largely untested.
- [W3] Many‑to‑many evaluation is qualitative only. This use‑case is shown via figures and a website but lacks quantitative measurements.
- [W4] Mixed performance is under‑discussed. FlowBind underperforms OmniFlow on some alignment metrics (e.g., T→I CLIP), while excelling on others (e.g., AIS, CLAP); the paper does not analyze why or when factorization helps/hurts. More explanations are needed.
- [W5] (Minor) Cost comparison caveat. The reported 10× compute reduction (48 vs 480 GPU‑hr) compares FlowBind’s full training to OmniFlow’s final joint stage only. Given the also‑much‑smaller data used by FlowBind, the speedup is promising but not an apples‑to‑apples pipeline cost.

**Questions:**

Please refer to weaknesses.

**Details Of Ethics Concerns:**

The system can synthesize convincing cross‑modal content (e.g., voice from images/text, imagery guided by audio/text) which raises impersonation and misinformation risks if deployed without safeguards. The paper does not discuss provenance, watermarking, or misuse mitigation; adding such guidance is advisable.

---

> ### Author Response · Authors · 2025-11-21
> **Rebuttal Response (1/4)**
>
> We thank the reviewer for the insightful and constructive suggestions on improvements. We address each of the reviewer’s comments below.
>
> ### **[W1] Ablation Studies on Shared-Latent Aggregation, Encoder/Decoder Choices**
>
> **On the Shared-latent Aggregation Rule**
>
> We thank the reviewer for this suggestion. In response, we conducted an ablation on the aggregation rule by using weighted averaging between the text and audio latents for the (Text+Audio) $\to$ Image generation task, evaluated on the AudioCaps test set. Specifically, at inference, we define the shared latent as:
>
>  $z^{*} = \lambda \cdot z_{\text{text}} + (1-\lambda) \cdot z_{\text{audio}}$.
>
> and vary $\lambda \in$ {$0.1, 0.3, 0.5, 0.7, 0.9$}. Note that $z_{\text{text}}$ and $ z_{\text{audio}}$ represent the latents encoded by each modality flow into the shared latent space. We then measure text-image alignment using CLIP score and audio-image alignment using AIS. The results are shown in Table A.
>
> **Table A. Weighted averaging of the shared latent for T + A  $\to$ I**
> | Metric | $\lambda= 0.1$ | $\lambda= 0.3$| $\lambda= 0.5$ | $\lambda= 0.7$ | $\lambda= 0.9$ |
> |--------------------|---------|---------|---------|---------|---------|
> | CLIP (T→I) | 23.30 | 25.04 | 25.57 | 25.92 | 27.73 |
> | AIS (A→I) | 59.59 | 59.17 | 56.89 | 56.09 | 55.18 |
>
>
> These results clearly show a trade-off: as the text weight $\lambda$ increases, text-image alignment improves, while audio-image alignment monotonically decreases. We empirically choose simple averaging ($\lambda = 0.5$) as our default since it offers balanced performance across different input modalities. This experiment demonstrates a practical advantage of our factorization: users can flexibly adjust $\lambda$ at inference time to prioritize the conditioning according to their specific intent.
>
> **Robustness Across Latent Feature Types**
>
> While designing the FlowBind to operate in the high-level semantic feature is one of our  contributions, FlowBind is not dependent on the specific choice of encoder/decoder. To validate this, we conducted ablation studies by substituting key components of the feature and encoder/decoder backbones, and then measured alignment performance across one-to-one generation tasks. In particular, we tried two variants: (1) replacing CLIP image features with DINO v2 ViT-Base/16 [1] features (using Lumos [2] as the decoder), and (2) replacing the Gemma3 text embeddings with OPTIMUS [3] (a text VAE) features. As summarized in Table B, FlowBind maintains consistent and reasonable performance under these substitutions, indicating robustness to different encoder/decoder backbone and latent representation.
>
>
> **Table B. FlowBind with different features**
> | Method | T$\to$I (CLIP) | I$\to$T (CLIP) | T$\to$A (CLAP) | A$\to$T (CLAP) | I$\to$A (AIS) | A$\to$I (AIS) |
>  |-------|------:|--:|---------------:|--------:|-----:|---:|
> | FlowBind | **28.35** | **29.74**| 29.08 | 36.70 | 82.89 | 78.17 |
> | FlowBind w/ DINOv2 [1] | 26.79 | 27.11 | 31.23 | **37.42** | 80.70 | **79.96** |
>  | FlowBind w/ Optimus [3] | 28.09 | 27.37 | **32.08** | 33.44 | **83.55** | 78.84 |
>
>
>
>
> [1] DINOv2: Learning Robust Visual Features without Supervision (TMLR 2024)
>
> [2] Learning Visual Generative Priors without Text (CVPR 2025)
>
> [3] Optimus: Organizing Sentences via Pre-trained Modeling of a Latent Space (EMNLP 2020)

---

> ### Author Response · Authors · 2025-11-21
> **Rebuttal Response (2/4)**
>
> ### **[W1-(iii), W2] Data-Flexibility Claim and Robustness Analysis with Varying Paired Data Ratios**
>
> We appreciate the reviewer’s comment. We first want to kindly remind the reviewer that our main experiment setting (Table 1 in the paper) leverages *only partially paired data*, such as Image-Text, Text-Audio, and Image-Audio pairs. In addition, Table 4 specifically targets an extreme, more challenging case: we train both FlowBind and the text-anchoring baseline using only Text–Image and Text–Audio pairs, and then evaluate them in a zero-shot manner on unseen Image–Audio translation. Although neither model has seen any Image–Audio pairs during training, FlowBind demonstrates strong performance on the I→A task in Table 4, with a significant gap compared to the text-anchored baseline. This directly supports our claim that FlowBind is data-flexible, demonstrating the ability to handle zero-shot, unseen paired cross-modal generation tasks.
> To further address the reviewer’s concern regarding the robustness of partial-paired data ratio, we comprehensively evaluate our method by varying the fraction of partially paired training data, extending the experiments from Table 4 of the main paper. Specifically, we vary the ratio of Image–Audio pairs used for training while keeping all other settings fixed. We subsample the Image–Audio pairs at different fractions (e.g., 0%, 1%, 3%, 10%, 30%, 100%) and measure alignment scores under the same conditions as in Table 4. The results are reported in Table C.
>
> **Table C. Varying Image-Audio Pair Ratios on Alignment Performance**
> | Method  | I→T  | A→T  | I→A  |
> |---|----|---|--|
> | Text anchoring  | 27.94 | 36.72 | 55.48 |
> | FlowBind 0% I-A  | 30.04 | 37.04 | 61.88 |
> | FlowBind 1% I-A  | 29.89 | 37.76 | 78.09 |
> | FlowBind 3% I-A |29.77 | 36.50 | 81.01 |
> | FlowBind 10% I-A| 29.77 | 36.17 | 82.98 |
> | FlowBind 30% I-A | 29.80 | 36.21 | 82.97 |
> | FlowBind 100% I-A  | 29.74 | 36.70 | 82.89 |
>
>
> These results demonstrate that FlowBind is robust to the ratio of partially paired data. Even with only 1% of the full Image–Audio dataset, which represents also a very small portion of the total data, FlowBind still achieves strong alignment. This suggests that FlowBind can maintain  reasonable performance across modalities, in scenarios with arbitrary partially paired data. Further details, including performance curves with respect to the varying data ratios and additional discussion, are provided in Figure 11 in Appendix G.

---

> ### Author Response · Authors · 2025-11-21
> **Rebuttal Response (3/4)**
>
> ### **[W3]  Lack of Many‑to‑Many Quantitative Evaluation**
>
> We would first like to note that, unlike paired data, collecting triplet (or higher-order) multimodal data is notoriously difficult, and such datasets are therefore scarce. For example, OmniFlow relies on synthetically generated triplets for training rather than naturally collected triplet data. Consequently, there has been no widely adopted standard benchmark for many-to-many generation tasks, which has made quantitative evaluation in this setting inherently challenging; prior works such as CoDi, OmniFlow, and LLM-based baselines have therefore focused primarily on qualitative demonstrations.
> We agree that many-to-many evaluation is important and that quantitative measurements are desirable. To resolve this concern, we constructed a synthetic triplet dataset extending the AudioCaps text–audio pairs. Following a protocol similar to OmniFlow, we generated the missing image modality using FLUX.1-schnell conditioned on the text annotations. This yields a triplet (text, audio, image) dataset that enables quantitative evaluation of many-to-many generation.
> The results are mentioned in Table D and Table E where we report the alignment scores for all input-output combinations in our many-to-one and one-to-many generation tasks.
>
> **Table D. Many-to-one generation alignment performances (Appendix Table 7)**
>
> | Method | (T+A) $\to$ I | | (T+I) $\to$ A | | (I+A) $\to$ T | |
> |---|-----|--|--|--|----|---|
>  | | CLIP (T$\to$I) | AIS (A$\to$I) | CLAP (T$\to$A) | AIS (I$\to$A) | CLIP (I$\to$T) | CLAP (A$\to$T) |
> | CoDi | 25.17 | 57.52 | 4.85 | 61.28 | 24.04 | 20.66 |
> | OmniFlow | 24.06 | 54.90 | 7.68 | 59.32 | 26.38 | 36.07 |
> | FlowBind | 25.57 | 57.93 | 28.13 | 76.02 | 27.83 | 35.21 |
>
>  **Table E. One-to-many generation alignment performances (Appendix Table 8)**
>
> | Method | T$\to$(I + A) | | I$\to$(T + A) | | A$\to$(T + I) | |
> |--|---|---|---|----|---|--|
> | | CLIP (T$\to$I) | CLAP (T$\to$A) | CLIP (I$\to$T) | AIS (I$\to$A) | CLAP (A$\to$T) | AIS (A$\to$I) |
> | CoDi | **26.61** | 10.99 | 25.73 | 58.65 | 18.03 | 57.14 |
> | OmniFlow | 24.71 | 12.92 | 26.36 | 63.99 | 36.07 | 54.22 |
> | FlowBind | 25.02 | **29.12** | **27.98** | **74.34** | **36.79** | **59.99** |
>
>
>
> As shown in the tables, FlowBind outperforms other flow-based models, achieving stronger alignment in both many-to-one and one-to-many generation tasks. In particular, for many-to-one generation, FlowBind better preserves information from both input modalities rather than effectively ignoring one of them. This highlights the effectiveness of our method in handling complex multimodal generation scenarios. We have also included detailed descriptions and quantitative results for these many-to-one and one-to-many settings in the appendix (Tables 7 and 8).
>
> Again, we agree that quantitative comparisons with baselines are important, and as such, we will move the tables and discussions from the many-to-many generation section in the appendix to the main paper in the final revision.
>
>
>
> ### **[W4] Analysis of Mixed Performance: Explaining the Strengths and Limitations of FlowBind with Respect to Factorization**
>
> As mentioned in L354-365, OmniFlow (and CoDi) leverage a specialized text-to-image diffusion model (e.g., SD3-Medium), optimized for T→I generation, resulting in strong T→I CLIP scores. FlowBind, in contrast, does not rely on a dedicated T→I backbone; instead, it learns all modality flows on top of a shared latent anchor, which explains the slight gap in T→I CLIP scores.
> Additionally, OmniFlow implicitly relies on text alignment, as it is initialized from pre-trained text-to-image and text-to-audio generative models. In contrast, FlowBind’s factorized shared latent is trained on arbitrary paired data (T–I, T–A, I–A), including direct image–audio pairs, in a manner symmetric across modalities. This design enables FlowBind to achieve significantly better AIS scores on image–audio generation tasks.
> In summary, while OmniFlow performs better on T→I due to its reliance on pre-trained specialist models, FlowBind’s factorized design allows it to effectively learn from arbitrary partially paired data.

---

> ### Author Response · Authors · 2025-11-21
> **Rebuttal Response (4/4)**
>
> ### **[W5] Cost comparison Caveat: Apples-to-Apple Comparison Between FlowBind and OmniFlow**
>
> Thank you for pointing this out. In practice, a fully controlled cost comparison is difficult because FlowBind and the baselines differ not only in training data scale, but also in model architectures and training protocols (e.g., single vs multi-stage training). Our intention was to provide a reference point by comparing our full training cost to OmniFlow’s final joint stage, which is the part of their pipeline most conceptually comparable to our cross-modal bridging setting. Given the many uncontrolled factors, we agree that this is not a perfectly apples-to-apples pipeline comparison, and we will clarify this limitation in the Section 5.1. If the reviewer has a preferred comparison protocol, we would be highly glad to follow that guidance and report the corresponding metrics.

---

### Author Response · Authors · 2025-11-21
**Revision Summary**

We sincerely thank the reviewer for their valuable feedback. Updates have been made in response to the reviewer’s feedback. The following changes have been implemented
- Updated Table 2 and 3 with more recent specialist models for comparison.
- **Appendix F**: Added quantitative results and analysis of many-to-one and one-to-many generation tasks, which will be moved to Section 5.2 in the final revision version of the paper.
- **Appendix G**: Included a brief analysis of data flexibility and the robustness of partially paired data ratios.
- **Appendix H**: Provided additional qualitative results on the robustness of plain averaging during inference time in the many-to-one generation task, with two conflicting inputs.
- **Appendix I**: Added figures and a discussion on the shared latent space of FlowBind, including t-SNE visualizations and corresponding generated images.
- **Appendix J**: Experiments with additional modality, 3D point clouds. Figure 14 (Image $\leftrightarrow$ Point clouds) and Figure 15 (Text $\leftrightarrow$ Point clouds) present the corresponding qualitative results.

---

### Meta-Review · Area_Chair_yEFK · 2026-01-07

**Summary:**

Across reviews, the submission is viewed as a clean, efficient any-to-any multimodal generation framework with a shared latent “anchor” and per-modality flows, and with generally competitive one-to-one results. The main drivers of uncertainty were (i) missing/insufficient ablations to support core claims (aggregation/fusion, backbone/latent choices, partial-pair ratios), (ii) lack of quantitative evaluation for many-to-many (or many-to-one / one-to-many) settings, (iii) under-analysis of mixed metric outcomes versus strong baselines, and (iv) questions about scalability beyond the three demonstrated modalities and about how much performance comes from frozen pretrained components rather than the proposed formulation. Overall, the rebuttal materially strengthens the empirical case on the key disputed points, shifting the balance toward acceptance.

**Reviewer Concerns:**

Addressed concerns:
- Authors added an explicit weighted-averaging study showing trade-offs and motivating simple averaging as a balanced default, plus backbone/feature substitutions indicating reasonable robustness. This directly targets the “insufficient ablations” concern and the fusion question.
- Authors added results varying paired ratios (including extreme low fractions) with clear improvements and robustness trends, addressing the “data-flexible” claim being previously under-tested.
- Authors constructed a triplet-style evaluation protocol and reported alignment numbers across input-output combinations with baseline comparisons, addressing the prior “qualitative only” concern.
- Authors clarified why some specialist-aligned metrics favor competing systems (strong pretrained specialist priors) while their method improves other cross-modal directions, partially addressing the “why/when it helps” critique.
- Authors added an additional modality experiment (point clouds) and qualitative unseen cross-modal combinations, addressing the scalability question at least preliminarily.

Remaining concerns:
- While formulation ablations help, one reviewer’s request for deeper explanatory analysis beyond empirical tables (mechanistic/diagnostic insight) is only partially satisfied.
- One reviewer asked for clearer runtime/steps/efficiency reporting; the rebuttal adds some runtime/NFE comparisons (at least for one direction), but broader coverage across tasks/modalities could still be stronger.
- Updated specialist comparisons help, but one reviewer’s broader concern about fully situating against the very latest systems across all tasks remains only partially resolved (the rebuttal adds several, but not exhaustive coverage).

**Reviewer Scores:**

- Reviewer kcBk: Likely upward from 6 → ~7 given direct additions on (1) aggregation ablations, (2) paired-ratio robustness, and (3) quantitative many-to-many-style evaluation; mixed-performance discussion was also addressed, though some “analysis depth” concerns may persist.
- Reviewer Hcpe: Already updated upward from 6 → 8 in the record after rebuttal; reviewer follow-up explicitly states no remaining concerns.
- Reviewer XtMR: Likely stable or slightly up from 8 → 8/9 since interpretability visualization and added-modality evidence were provided; their main remaining critique (deeper “why” analysis) is improved but may not fully warrant a jump.
- Reviewer 1rzN: Likely upward from 4 → ~5/6 because the rebuttal directly targets (i) outdated baselines (added newer specialists), (ii) confounding by frozen models (formulation ablations + alternate backbones), (iii) many-to-one/one-to-many quantification, and (iv) extra modality scaling; however, their original skepticism about competitiveness vs the latest ecosystem may not be fully eliminated.

---

### Decision · Program_Chairs · 2026-01-26

Accept (Poster)